# DNA Language Models for mRNA Analyses

## Abstract

Genomic Language Models (gLMs), encompassing DNA models, RNA models, and multimodal models, are becoming widely used for the analysis of biological sequences. Typically, models trained on RNA are used for RNA-related tasks, and models trained on DNA sequences are used for DNA tasks. However, this requires the development and maintenance of several classes of models to match the modality of the sequence. These models take significant resources and data to create, and maintaining separate models for DNA and RNA tasks is a computational burden.

To reduce this burden, we introduce novel Adaptive Mixture of Codon Reformative Experts (CodonMoE) that can be incorporated into DNA gLMs in order to adapt them for mRNA-based predictive tasks. We show that, by using this plug-and-play operator, DNA-based gLMs can achieve performance similar to that of RNA-trained models on mRNA tasks. We further show that recent, efficient subquadratic DNA-based state space model (SSM) architectures can be used with the CodonMoE to achieve parameter- and computationally-efficient predictions for mRNA tasks. Specifically, experimental results demonstrate that CodonMoE improves diverse DNA-based backbones by a large margin, with some models achieving comparable or superior performance to current state-of-the-art RNA-specific models across several downstream tasks, while reducing both time complexity and model parameters.

Our results provide a path for focusing development efforts of gLMs on DNA models, which can then be adapted to mRNA tasks. Because DNA data is more prevalent than assembled mRNA data, and modeling efforts can focus on a single class of model, this is likely to foster improved DNA models for mRNA tasks at lower computational cost and is a significant step towards unifying genomic language modeling.

## 1 Introduction

Recent advancements in artificial intelligence, particularly in the domain of Large Language Models (LLMs), are revolutionizing numerous scientific disciplines, with the biomedical sciences experiencing especially profound impacts (Jumper et al., 2021; Varadi et al., 2022). The fundamental goal of Natural Language Processing (NLP) is to comprehend and manipulate sequences of words, a task that bears similarities to one of the central objectives in biology: deciphering the meaning and function encoded in biological sequences (Eraslan et al., 2019), as well as designing and generating novel genomic sequences with desired properties. This parallel has given rise to a new frontier in computational biology: Genomic Language Models (gLMs). GLMs are large-scale language models trained on vast amounts of biological sequence data. These models aim to capture the complex patterns and dependencies within genomic sequences, much like how general LLMs learn the intricacies of human language (Bepler & Berger, 2021). By leveraging the power of large language models and the abundance of genomic data now available, gLMs have the potential to significantly advance our understanding of genomes and reveal how DNA or RNA elements at various scales interact to give rise to biological functions (Zhou et al., 2018).

Recent progress in state-space models (SSMs) have addressed the quadratic scaling limitations inherent in self-attention mechanisms, offering efficient alternatives to transformers for gLMs (Ji et al., 2021; Benegas et al., 2023; Ratcliff, 2024) with subquadratic or linear scaling in sequence length. HyenaDNA (Nguyen et al., 2024b), built on the Hyena Hierarchy, represents a significant leap for-

ward in genomic modeling, processing input contexts up to 1 million nucleotides — a 500-fold increase over previous dense attention-based models. This architecture enables single-nucleotide-level analysis across extensive genomic regions, crucial for capturing long-range interactions and subtle genetic variations like SNPs. Caduceus (Schiff et al., 2024), leveraging the Mamba-based SSM (Gu & Dao, 2023), introduces bi-directionality and reverse complementarity (RC) equivariance, essential properties for comprehensive DNA sequence analysis. Trained on 131 kb sequences, Caduceus demonstrates superior performance on long-range prediction of variant effects tasks compared to much larger models. Building upon this framework, PlantCaduceus (Zhai et al., 2024) extends these capabilities to diverse plant genomes, showcasing high transferability across species that diverged 160 million years ago and enabling genome-wide deleterious mutation identification without multiple sequence alignment. EVO (Nguyen et al., 2024a), a hybrid architecture combining Hyena and Transformer elements, pushes the boundaries further with its 7 billion parameter model and 131 kb context length. EVO's multi-modal approach allows it to generalize across DNA, RNA, and protein prediction tasks, while also demonstrating unprecedented capabilities in generating synthetic molecular complexes and coding-rich sequences up to 650 kb in length.

Despite significant advancements in genomic language modeling, the development of distinct gLMs—encompassing DNA models, RNA models, and multimodal models—introduces a considerable cost burden. This issue becomes increasingly pronounced as the size and complexity of gLMs grow. Moreover, attention-based models, particularly in the context of RNA language modeling, continue to dominate most RNA-specific tasks. Although these models deliver strong performance, their high computational demands remain a substantial challenge. According to the central dogma of molecular biology, DNA serves as the primary repository of genetic information, while mRNA functions as an intermediary in the expression of this information (Crick, 1970). Building upon this fundamental concept, DNA-based language models offer a more holistic and foundational approach to genomic modeling compared to mRNA-focused models. However, despite their great potential, DNA-based models have largely been underutilized in downstream mRNA analyses.

To address these challenges, we propose a novel approach based on the hypothesis that DNA models can effectively replace RNA models when augmented with RNA-specific control information. Central to our method is **Adaptive Mixture of Codon Reformative Experts (CodonMoE)**, a versatile plug-and-play module designed to seamlessly integrate with existing DNA models, transforming them into robust tools for mRNA analyses. We also demonstrate that recent, efficient sub-quadratic DNA-based state space model (SSM) architectures can be effectively combined with the CodonMoE to yield parameter- and computationally-efficient predictions for mRNA tasks. This marks the first approach to bridge the gap between DNA and RNA language models through a universally applicable CodonMoE.

Theoretical proof demonstrates that CodonMoE is a universal approximator of mRNA properties at the codon level. Experimental results further show that CodonMoE significantly enhances various DNA-based backbones by a wide margin, as illustrated in Figure 1. Some of these models achieve performance comparable to or exceeding state-of-the-art (SOTA) mRNA-specific models across critical downstream tasks, while also achieving substantial reductions in time complexity and model parameters.

In general, CodonMoE offers the following "3A" characteristics in versatility:

- **Adaptability**: CodonMoE integrates seamlessly with a variety of DNA model architectures, including SSMs and attention-based models, ensuring compatibility across diverse computational frameworks.

- **Applicability**: CodonMoE is capable of handling DNA models trained on datasets from diverse species, making it suitable for a wide range of biological tasks without being restricted by species-specific data.

- **Across-Species Generalization**: CodonMoE consistently enhances DNA models for mRNA-related tasks, achieving high performance even when applied to species not represented in the original training data, thereby demonstrating broad utility across multiple species in RNA analyses.

Source code for this work is available at https://anonymous.4open.science/r/CodonMoE.

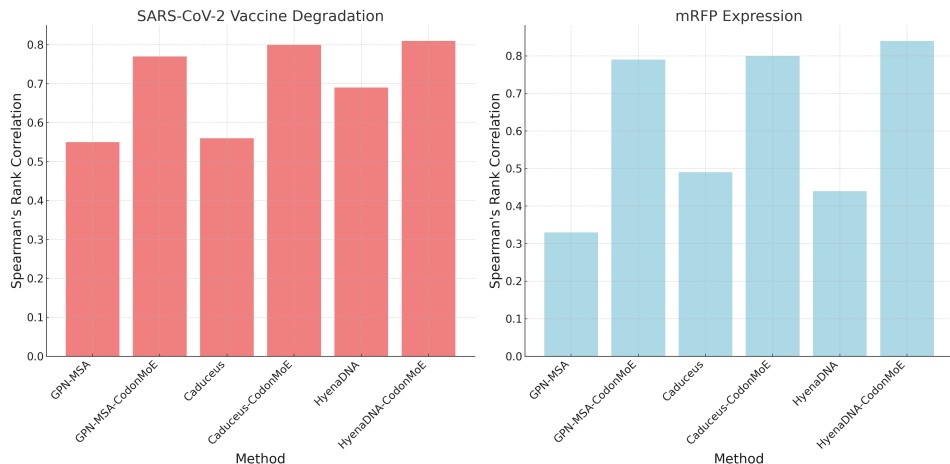

Figure 1: Performance comparison on mRFP expression and SARS-CoV-2 vaccine degradation datasets across GPN-MSA (Benegas et al., 2023), HyenaDNA (Nguyen et al., 2024b), and Caduceus (Schiff et al., 2024) models, with and without our CodonMoE integration.

## 2 RELATED WORK

**Transformer-based genomic language models.** Transformer models (Vaswani et al., 2017) (Devlin, 2018) have become a popular choice for genomics modeling, offering the ability to capture long-range dependencies critical for DNA and RNA sequence analysis (Benegas et al., 2024). Despite their success, transformer-based models often face limitations in handling long context lengths and relying on tokenization schemes that aggregate nucleotides into basic language model units, compromising single-nucleotide resolution. In the DNA space, DNABERT (Ji et al., 2021) tackles tasks like transcription factor binding site prediction by adapting the BERT architecture with DNA tokenized with k-mer, demonstrating the potential of transformers to capture long-range dependencies in genomic data. Enformer (Avsec et al., 2021) further extends this concept by incorporating convolution layers before and after transformer blocks. Nucleotide Transformer further pushes the boundaries of what transformers can achieve in genomics, achieving five times the scale of DNABERT and ten times that of Enformer (Dalla-Torre et al., 2023). MegaDNA (Shao, 2023), a multiscale transformer model for bacteriophage genomes, extends the context window to accommodate longer sequences, and showcases the potential of transformers in generative tasks. GPN-MSA (Benegas et al., 2023), unlike these models, offers an approach leveraging whole-genome sequence alignments across multiple species, demonstrating how the evolutionary structure of sequences enhances DNA modeling tasks.

On the RNA side, transformer-based models like RNABERT (Akiyama & Sakakibara, 2022) and BigRNA (Celaj et al., 2023) have also been developed to address various transcriptomic tasks. Specialized models like CodonBERT (Li et al., 2024) and SpliceBERT (Chen et al., 2023) focus on tasks like codon-level translation and splicing, respectively, while scBERT (Yang et al., 2022) targets single-cell RNA-seq data annotation. Despite these advancements, RNA transformer models share similar challenges to their DNA counterparts, particularly when handling long sequences and maintaining computational efficiency. These limitations have fueled the rise of state-space models (SSMs) as an alternative, offering reduced time complexity and improved scalability for long-range dependencies.

**SSM-based genomic language models.** In response to the limitations of transformers, state-space models (SSMs) have gained traction in genomic language modeling, offering the ability to handle longer context lengths with reduced time complexity. Models such as HyenaDNA (Nguyen et al., 2024b) and Caduceus (Schiff et al., 2024) have proven effective in sequence modeling tasks, capitalizing on the strengths of SSMs. Going beyond sequence modeling, EVO (Nguyen et al., 2024a) showcases the potential of an SSM-based model for whole-genome-scale DNA generation. These developments underscore the increasing significance of SSMs in genomic research, offering pow-

erful tools for large-scale sequence analysis and generation. In addition to DNA, EVO has learned information encoded in other modalities including RNA.

**Mixture of Experts.** Mixture of Experts (MoE) enhances model performance by a set of experts focusing on different aspects of input data. The concept was first introduced in Jacobs et al. (1991), and extended to hierarchical settings in Jordan & Jacobs (1994). In the Natural Language Processing (NLP) domain, Shazeer et al. (2017) introduced a sparsely-gated MoE layer, with only a subset of experts activated for each input, thereby improving efficiency and scalability, successfully scaled MoE to a 137 billion parameter LSTM. Building on this idea, Lepikhin et al. (2021) scaled up transformers beyond 600 billion parameters with GShard, demonstrating MoE's effectiveness in large-scale models. The Switch Transformer (Fedus et al., 2021) simplified gating to select a single expert, leading to a 1.6T-parameter MoE. Additionally, GLaM (Du et al., 2021) uses sparse activation to further scale up, matching GPT-3 quality with only one-third of the energy. Finally, Zuo et al. (2022) refined the activation process proposed with stochastic experts, enhancing sparsity management. In summary, MoE enhances model performance by dynamically activating the most relevant experts and allows efficient scaling of models and datasets while reducing computational effort.

## 3 METHODOLOGY

We introduce a novel module CodonMoE that can be integrated into state-of-the-art pretrained SSMs and attention-based models designed for DNA sequence analysis for adapting them for RNA analyses. The CodonMoE processes these hidden states from those DNA backbones by restructuring the input into codons (three-nucleotide sequences) and applying an Adaptive Mixture of Codon Reformative Experts. Each expert within the CodonMoE is designed to identify and emphasize various biological signals, enabling the model to capture both codon-level and broader sequence patterns. Furthermore, we demonstrate that the CodonMoE is a universal approximator at the codon level. Given sufficient expert capacity, the CodonMoE can approximate any continuous function that maps codon sequences to specific target properties with arbitrary precision when combined with the pretrained backbone model. In general, this architecture effectively translates DNA models to RNA contexts, allowing for robust analysis of RNA sequences.

### 3.1 MODEL OVERVIEW

As illustrated in Figure 2, our architecture is underpinned by a state-of-the-art, pretrained state space model (SSM) originally designed for DNA language analysis. This robust framework is augmented by the CodonMoE, a modular enhancement specifically developed to translate DNA-centric models for RNA sequence analysis. The base model extracts hidden states, capturing patterns encoded within DNA sequences. These states are subsequently processed by the CodonMoE, which employs a novel approach to adapt these DNA-derived patterns for mRNA contexts. This adaptation process begins with the grouping of inputs into codons which is followed by the deployment of our Adaptive Mixture of Codon Reformative Experts, each expert fine-tuned to recognize and amplify different biological signals inherent in the sequence data. This model will be introduced in detail in this section.

### 3.2 CODONMOE: ADAPTIVE MIXTURE OF CODON REFORMATIVE EXPERTS

**Sample-wise dynamic codon-level representation.** The CodonMoE processes representations of codons, which are groups of three nucleotides in genetic sequences encoding amino acids. The input to CodonMoE consists of nucleotide representations with dynamic dimensionality, allowing it to accommodate input samples of varying sequence lengths. These inputs are reshaped into codon groups, preserving the structure of the genetic code is preserved. The CodonMoE slices this sequence to extract codon-related segments and reshapes them to facilitate further processing.

**Adaptive Mixture of Reformative Codon Experts.** One of the core CodonMoE functionalities is handled by Adaptive Mixture of Codon Experts layers, where multiple experts, each specializing in different aspects of the codon data, process these representations. The transformation is given by:

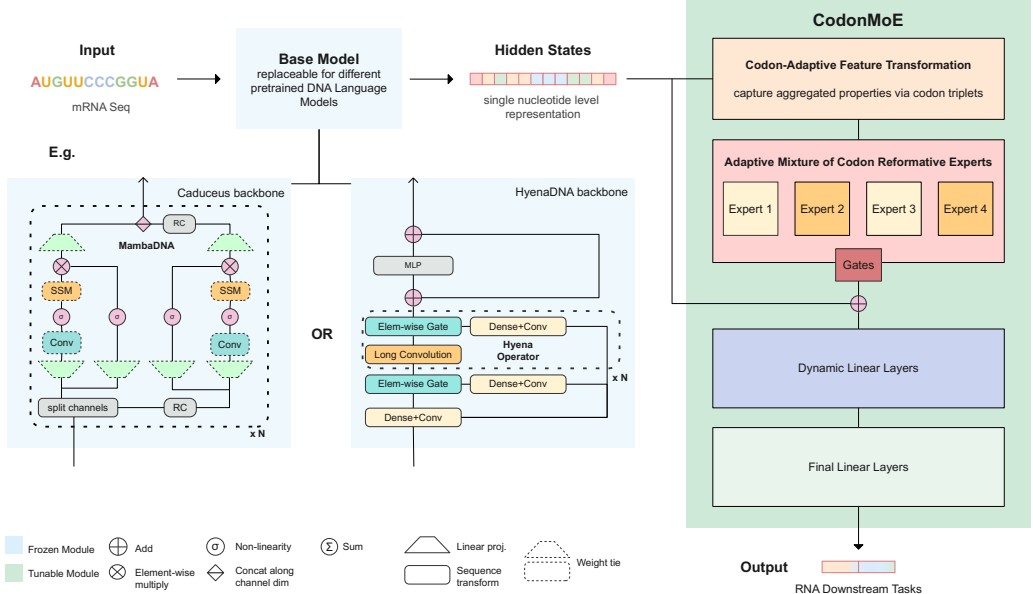

Figure 2: Overview of CodonMoE and proposed framework. The architecture combines pretrained DNA-focused state space models with a novel CodonMoE module. This CodonMoE adapts DNA-derived patterns for RNA analysis by grouping inputs into codons and using a Mixture of Experts (MoE) approach. This design enables effective translation of DNA models for RNA sequence analysis, leveraging the strengths of both domains.

$$y_{\text{codons}}^{\text{MoE}} = \sum_{k=1}^{K} g_k(x) E_k(y_{\text{codons}}),$$

where $g_k(x)$ is the gating mechanism that determines the contribution of each expert $E_k$. This dynamic expert selection allows the MoE to process the codon data in multiple ways, with the gating system controlling which perspective should dominate.

**Dynamic reshaping and contextualization.** After processing by the experts, the codon-level representations are reshaped to match the original sequence length and structure. The CodonMoE contextualizes this information, enriching it with surrounding data before recombining it with the rest of the input sequence:

$$y_{\text{output}} = y_{\text{reshaped}} + y_{\text{codons}}^{\text{MoE}}.$$

This process ensures that codon-level information is properly embedded and aligned within the original sequence, helping the model recognize both local codon-specific patterns and broader genetic patterns.

For more detailed specification of the algorithm, please refer to the appendix A.1.

### 3.3 CODONMOE IS A UNIVERSAL APPROXIMATOR AT CODON LEVEL

We show that given sufficient capacity, our proposed CodonMoE can approximate any function that maps codon sequences to target properties with arbitrary precision when integrated with the pretrained backbone model.

**Definitions and Preliminaries** We begin by defining the key concepts.

**DNA Sequence Space** ($\mathcal{X}$) is defined as the set of all possible DNA sequences composed of nucleotides from the alphabet $\{A, C, G, T\}$. In our framework, the RNA nucleotide 'U' is systematically replaced with the DNA nucleotide 'T', aligning RNA codons with their corresponding DNA

representations. This substitution ensures compatibility between RNA and DNA sequences within our model. **Codon Space** ($\mathcal{C}$) consists of all possible codons, where each codon is a sequence of three nucleotides from $\{A, C, G, T\}$. Formally, $\mathcal{C} = \{A, C, G, T\}^3$. **Function Class** ($\mathcal{F}$) comprises all continuous functions $f : \mathcal{C}^n \to \mathbb{R}$ that map sequences of $n$ codons to specific target properties, where $n$ is the number of codons in the sequence.

Our modeling approach is structured around a two-stage paradigm. Initially, a **Backbone Model** $h : \mathcal{X} \to \mathbb{R}^{L \times D}$ is pretrained on DNA sequences, where $L$ represents the sequence length and $D$ the embedding dimension. This pretraining phase equips the backbone with foundational knowledge of genetic sequences and their inherent patterns. We directly use the pretrained models on DNA sequences. Subsequently, the **CodonMoE** serves as an adapter to this pretrained backbone model. Formally, the CodonMoE is a function $g : \mathbb{R}^{L \times D} \to \mathbb{R}$ that is fine-tuned on mRNA sequences to specialize the model for mRNA-specific tasks. This fine-tuning process involves training the adapter using mRNA sequences, which have been converted by replacing 'U' with 'T', thereby maintaining consistency with the DNA-based backbone.

**Theorem 3.3** Let $\mathcal{C} = \{A, C, G, T\}^3$ be the codon space, and let $\mathcal{F} = \{f : \mathcal{C}^n \to \mathbb{R} \mid f$ is continuous$\}$ be the class of target functions. Consider a pretrained backbone model $h : \mathcal{X} \to \mathbb{R}^{L \times D}$, where $\mathcal{X} = \{A, C, G, T\}^*$, and an adapter CodonMoE $g : \mathbb{R}^{L \times D} \to \mathbb{R}$ structured as a dense MoE with $K$ experts. Assume the following conditions hold:

1. **Expert Capacity**: Each expert $E_k : \mathbb{R}^D \to \mathbb{R}^{D'}$ within the MoE is a neural network capable of uniformly approximating any continuous function on compact subsets of $\mathbb{R}^D$.

2. **Gating Mechanism**: The gating network $G : \mathbb{R}^D \to \Delta^K$ (where $\Delta^K$ is the $K$-simplex) assigns non-negative weights $g_k(z_i)$ to each expert based on the input $z_i \in \mathbb{R}^D$, satisfying $\sum_{k=1}^{K} g_k(z_i) = 1$.

3. **Embedding Representation**: Each DNA sequence $x \in \mathcal{X}$ is partitioned into codons $(c_1, c_2, \ldots, c_n)$, and the backbone model generates embeddings $h(x) \in \mathbb{R}^{L \times D}$, where $L = 3n$ (assuming each codon is represented by three consecutive embeddings).

Then, for any function $f \in \mathcal{F}$ and for any $\epsilon > 0$, there exists a number of experts $K$ and corresponding parameters for the CodonMoE such that, for all $x \in \mathcal{C}^n$, the approximation error satisfies

$$\left| f(c_1, c_2, \ldots, c_n) - g\left( \sum_{i=1}^{n} \sum_{k=1}^{K} g_k(z_i) \cdot E_k(z_i) \right) \right| < \epsilon,$$

where $z_i = [h(c_i)] \in \mathbb{R}^D$ is codon $c_i$ represented by averaging three nucleotide embeddings.

For technical proof of Theorem 3.3, please refer to the appendix A.2.

# 4 EXPERIMENTS

## 4.1 TASKS AND DATASETS

**mRFP expression dataset.** We have used the monomeric Red Fluorescent Protein (mRFP) expression dataset generated by Nieuwkoop et al. (2023). This dataset consists of 1,459 unique mRFP variants, each with paired expression levels (the target variable) and sequence data. These variants are derived from three codon-randomized libraries with varying codon adaptation index (CAI) biases, allowing for analysis of how sequence variations impact mRFP expression.

**SARS-Cov-2 vaccine degradation dataset.** For our analysis of mRNA design principles for SARS-CoV-2 vaccines, we have used the comprehensive dataset generated by Leppek et al. (2022). This dataset contains 2,400 samples, with each sample including data on vaccine stability or degradation (the target variable) and associated sequence characteristics, providing insight into factors affecting mRNA vaccine durability.

Both datasets were selected to evaluate the performance of our models and are the same as several used in the CodonBERT paper (Li et al., 2024), facilitating direct comparison across key mRNA-related prediction tasks.

For more details of the datasets, please refer to the appendix section A.3.

## 4.2 Experimental Settings

For each dataset, two SSM-based backbones including Caduceus and HyenaDNA and one attention-based backbone GPN-MSA are tested with different variants of CodonOperator. The baseline experiments for DNA backbone feature analysis use various regressors such as MLP and XGBoost, with specified learning rates and epochs for certain models. For more detailed experimental configurations and parameters, please refer to the appendix section A.3 and A.4.

## 4.3 Main Results

The table presented (Table 1) offers comparisons of state-of-the-art codon-based RNA and DNA language models, with a specific focus on enhancements from both computational cost and performance aspects provided by the CodonMoE. Metrics for evaluation include the Spearman's rank Correlation for the SARS-CoV-2 vaccine degradation and mRFP expression datasets, which measures the models' ability to accurately capture and predict biologically relevant patterns. A high Spearman's rank correlation indicates that the model effectively ranks biological variables in alignment with experimental observations, thus validating its predictive power in complex biological processes.

The CodonMoE's integration into existing DNA models demonstrates marked improvements in mRNA analyses, as indicated by Spearman's rank correlation metrics. The integration of the Codon-MoE transforms diverse DNA models into significantly more powerful tools for mRNA analysis. This is evident from the performance leaps observed in models like HyenaDNA-CodonMoE and Caduceus-CodonMoE, where the CodonMoE not only amplifies their inherent capabilities but also enables them to rival or surpass state-of-art codon-based RNA models in performance with much fewer model parameters, which are reduced by above $80\%$ compared with attention-based mRNA specific state-of-art models.

### 4.3.1 RNA-Based Benchmark

The authors of CodonBERT (Li et al., 2024) evaluated three prominent RNA-based models, each of which exhibits quadratic time complexity due to their reliance on self-attention mechanisms. The first model, RNABERT + TextCNN (Akiyama & Sakakibara, 2022; Li et al., 2024), integrates a pretrained RNABERT architecture with a TextCNN layer tailored for downstream tasks. Despite having fewer than 20 million parameters, this model demonstrated competitive performance in both RNA-related tasks.

In contrast, RNA-FM + TextCNN (Chen et al., 2022; Li et al., 2024), with over 80 million parameters, leverages a larger architecture combining RNA-FM pretraining with a TextCNN layer. This more extensive architecture demonstrated an enhanced capacity for sequence feature extraction, performing better in tasks requiring greater complexity.

Finally, CodonBERT (Li et al., 2024), specifically optimized for codon-based RNA tasks, emerged as the top-performing RNA language model among our baselines. This model's fine-grained understanding of codon patterns positions it as the leading benchmark for RNA-specific downstream tasks, though it has quadratic time complexity and a large parameter count.

### 4.3.2 CodonMoE Leads to Computational Efficiency and Performance Superiority over DNA Models

**Base models and enhanced models with CodonMoE.** The GPN-MSA (Benegas et al., 2023) and Caduceus (Schiff et al., 2024) models, in their standard configurations without the CodonMoE enhancements, exhibit moderate-to-low performance metrics. Specifically, the Caduceus model shows a notable underperformance in predicting SARS-CoV-2 vaccine degradation outcomes. Integration of the CodonMoE significantly improves both models. GPN-CodonMoE and Caduceus-CodonMoE display substantial improvements in their Spearman scores, illustrating the CodonMoE's efficacy in enhancing the capabilities of DNA-based models. The HyenaDNA model (Nguyen et al., 2024b) exhibits variable outcomes in its standard and enhanced forms. The integration of the CodonMoE

(HyenaDNA-CodonMoE) markedly boosts its performance, achieving the highest Spearman correlations in the group. This significant enhancement in processing mRNA sequences underscores the computational efficiency impact of the framework, which includes our CodonMoE.

**Computational efficiency and parameter efficiency.** Both the Caduceus and HyenaDNA models, even when augmented with the CodonMoE, maintain a linear or subquadratic time complexity. This characteristic is highly advantageous, enabling the efficient processing of extensive genomic datasets. Enhanced models, such as Caduceus-CodonMoE and HyenaDNA-CodonMoE, not only perform well but also maintain a minimal parameter footprint, with fewer than 20 million parameters. This efficiency highlights their potential for scalable deployment in diverse genomic applications.

Based on the aforementioned findings, we infer that CodonMoE-augmented models benefit from efficient codon-level embeddings, which allow the models to capture the functional differences between codons and their impact on mRNA properties. This enables the model to predict which sequences are optimal for high protein expression. The models efficiently acquire knowledge regarding the contextual interaction of codons within a larger mRNA sequence, with the support of SSM architectures. This is crucial because the secondary structure of the mRNA can be influenced by the modification of a single codon, which in turn affects the stability and translation of the mRNA.

Meanwhile, as indicated in Table 1, SSMs are designed to handle long sequences, making them ideal for processing the long contexts required to model codon interactions effectively. This is critical for understanding the secondary structure of mRNA, where codon interactions over long distances significantly influence folding and stability. The ability of SSMs to capture these dependencies efficiently provides a substantial edge over traditional models, which often struggle with computational costs and context limitations in long-range sequence tasks.

Table 1: Evaluation of computational complexity and Spearman's rank correlation metrics across RNA and DNA language models: delineating the impact of CodonMoE integration on model performance and parameter efficiency. CodonMoE suffix indicates models enhanced with our proposed CodonMoE module. Each data set is split into training, validation, and testing with a 0.7, 0.15, and 0.15 ratio, using the same split set as in the CodonBERT (Li et al., 2024). The metric is Spearman's rank Correlation.

| Method | Modality | Time Complexity | Model Parameters | Vaccine Degradation | mRFP Expression |
|---|---|---|---|---|---|
| | | RNA Models | | | |
| RNABERT$_{+TextCNN}$ | RNA | quadratic | <20M | 0.64 | 0.40 |
| RNA-FM$_{+TextCNN}$ | RNA | quadratic | >80M | 0.74 | 0.80 |
| CodonBERT | RNA | quadratic | >80M | **0.77** | **0.85** |
| | | DNA Models | | | |
| GPN-MSA | DNA | quadratic | >80M | 0.55 | 0.33 |
| GPN-MSA-CodonMoE | DNA | quadratic | >80M | 0.77 | 0.79 |
| Caduceus | DNA | linear | <20M | 0.56 | 0.49 |
| Caduceus-CodonMoE | DNA | linear | <20M | 0.80 | 0.80 |
| HyenaDNA | DNA | subquadratic | <20M | 0.69 | 0.44 |
| HyenaDNA-CodonMoE | DNA | subquadratic | <20M | **0.81** | **0.84** |

## 4.4 ABLATION STUDY

### 4.4.1 COMPARATIVE ANALYSIS OF CODONOPERATOR VARIANTS IN RNA MODELING

To test whether more complex frameworks like the MoE are necessary, we implemented and evaluated a simpler approach. We developed a method called CodonMean, which computes the mean of codon features derived from three nucleotide embeddings extracted from the backbone models. This method acted as a lightweight and parameter-efficient adapter. While CodonMean yielded improvements on key mRNA tasks compared to using pure DNA backbones, it struggled to reach the performance levels of existing codon-based RNA models that typically leverage attention-based mechanisms. This led us to explore more sophisticated approaches, ultimately resulting in the development of CodonMoE as a more advanced and effective solution.

For the mRFP expression task, the experiments were conducted on three different DNA models: GPN-MSA, HyenaDNA, and Caduceus, with two versions of the CodonOperator: **CodonMean** and **CodonMoE**. As shown in Table 3, the integration of either codon operator significantly improved the performance of all these DNA models. CodonMean, which employs a simple codon-mean aggregation, produced strong results. CodonMoE, which uses a more sophisticated Mixture of Experts (MoE) mechanism to better capture codon-level dependencies, outperformed the Codon-Mean across all models.

Table 2: CodonOperator variant comparison on mRFP expression dataset.

|  | GPN-MSA | HyenaDNA | Caduceus |
|---|---|---|---|
| CodonMean | 0.740 | 0.765 | 0.766 |
| CodonMoE | **0.790** | **0.837** | **0.802** |

In the SARS-CoV-2 vaccine degradation task, we further validated the applicability of codon operators in enabling DNA models to perform well in mRNA-focused tasks. As with the mRFP task, both codon operators versions were tested across GPN-MSA, HyenaDNA, and Caduceus models (Table 3). CodonMean delivered a solid performance. However, CodonMoE once again showed its superiority, achieving the highest scores across all models.

Table 3: CodonOperator variant comparison on SARS-CoV-2 vaccine degradation dataset.

|  | GPN-MSA | HyenaDNA | Caduceus |
|---|---|---|---|
| CodonMean | 0.729 | 0.789 | 0.755 |
| CodonMoE | **0.770** | **0.812** | **0.795** |

The results from both tasks underscore the flexibility and impact of a codon operator. As a plug-and-play module, CodonOperator can be integrated into nucleotide-level DNA models, enabling it to effectively handle RNA downstream tasks. This approach not only enhances the predictive power of DNA models but also brings them to the forefront of RNA-specific challenges.

### 4.4.2 EVALUATING THE EFFECTIVENESS OF PRETRAINED DNA MODEL FEATURES FOR RNA TASKS

To further investigate the effectiveness of the features extracted by prevailing DNA models for RNA-related tasks without using a codon operator, we conducted ablation studies using two regression methods: MLP and XGBoost. These models were applied to features directly extracted from pre-trained GPN-MSA, HyenaDNA, and Caduceus models that were not augmented with any codon operator. The goal of this ablation study was to evaluate how well the raw features from the pretrained models perform in downstream tasks when processed by external regression models, as opposed to using our tunable CodonMoE integrated into diverse nucleotide-level DNA backbones.

In the mRFP expression task presented in Table 4, we extracted features from the pretrained GPN-MSA, HyenaDNA, and Caduceus models and applied them to both MLP and XGBoost models. The results indicated that XGBoost was more effective when using features from GPN-MSA and Hye-naDNA, where it demonstrated bet-

Table 4: Evaluation of DNA pretrained model feature effectiveness on mRFP expression dataset using MLP and XG-Boost.

|  | GPN-MSA | HyenaDNA | Caduceus |
|---|---|---|---|
| MLP | 0.330 | 0.439 | **0.490** |
| XGBoost | **0.479** | **0.512** | 0.476 |

ter performance overall, which showed a stronger capability in handling the features extracted from HyenaDNA, suggesting that its more complex, decision tree-based architecture is better aligned with the structure of HyenaDNA's feature representations. For the SARS-CoV-2 vaccine degradation task, XGBoost consistently outperformed MLP across all three models as shown in Table 5. This indicates that XGBoost's ability to handle complex interactions between features made it more suitable for this particular task. MLP, while performing reasonably well with HyenaDNA, was less effective with the features extracted from GPN-MSA and Caduceus.

**Analysis.** These ablation studies reveal the strength and limitations of the feature representations learned by the pretrained DNA models for mRNA tasks. While MLP exhibits some capability to process these features, particularly for GPN-MSA and Caduceus in the mRFP expression task, XGBoost generally performed

Table 5: Evaluation of DNA pretrained model feature effectiveness on SARS-CoV-2 vaccine degradation dataset using MLP and XGBoost.

|         | GPN-MSA | HyenaDNA | Caduceus |
|---------|---------|----------|----------|
| MLP     | 0.572   | 0.695    | 0.560    |
| XGBoost | **0.750** | **0.711** | **0.737** |

better, especially in the SARS-CoV-2 degradation task. This supports the idea that XGBoost's tree-based architecture is better suited for handling the structured and possibly sparse features generated by DNA-pretrained models, offering more stable and higher performance without requiring extensive tuning. The results can be firstly attributed to XGBoost being more robust and less sensitive to hyperparameter tuning compared to MLPs, which require careful optimization of neural network parameters for optimal performance. Secondly, while the raw features from pretrained DNA models contain some information about RNA, directly applying DNA models to mRNA analyses is suboptimal for downstream tasks (compare Table 4 and Table 5 with Table 1). This is partly because DNA models have not been trained to capture mRNA-specific properties, instead focusing on more fundamental nucleotide characteristics and DNA-specific interactions and functions.

### 4.4.3 CONSISTENT PERFORMANCE OF CODONMOE ACROSS DIFFERENT MODELS

Integrating the CodonMoE module into the GPN-MSA, HyenaDNA, and Caduceus models resulted in significant performance improvements across critical genomic prediction tasks as presented in Figure 1. Additionally, the results indicate that standard DNA models perform poorly on mRNA tasks, which is expected since these models are pretrained on DNA data and capture sequence properties distinct from mRNA. However, with our proposed CodonMoE, a codon-aware, plug-and-play module, the performance of the models consistently improves by a significant margin. This highlights the effectiveness of codon-based adapters, which not only leverage the rich information within DNA models but also enhance mRNA analysis capabilities. In all cases, the models exhibited enhanced accuracy in predicting mRNA expression levels and vaccine degradation. Moreover, the feature visualization comparisons between the backbones with and without CodonMoE align closely with the results presented in Figure 1. For a more detailed discussion of these visualization comparisons and more experiments, please refer to Appendix A.5 and Appendix A.6.

## 5 CONCLUSION

Our theoretical and experimental results highlight the characteristics of CodonMoE. Firstly, CodonMoE is highly adaptable to various DNA model architectures, such as state space models (SSMs) and attention-based models, providing flexibility across different computational frameworks. Moreover, it is also applicable to DNA models trained on datasets from diverse species, making it well-suited for generalized biological contexts without being restricted to species-specific data. Furthermore, CodonMoE performs well in mRNA-related tasks, significantly enhancing the performance of DNA backbones and providing comparable or even superior performance to RNA-specific models across several downstream tasks, while reducing computational burden. Its versatility allows it to maintain high performance even when applied to species not present in the DNA model training dataset, offering broad utility across multiple species in mRNA analyses.

Our findings delineate an approach for directing the formation of gLMs toward DNA models, which can then be modified for mRNA applications. The predominance of DNA data over assembled mRNA data, coupled with the ability to concentrate modeling efforts on a single model class is expected to enhance DNA models for mRNA tasks at reduced computational expense, representing a crucial advancement in the unification of genomic language modeling.

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

# A APPENDIX

## A.1 ALGORITHM PSEUDOCODE

The proposed CodonMoE whose pseudocode is given in Algorithm 1, efficiently analyzes mRNA sequences by leveraging a novel MoE model tailored for codon-level feature extraction. This method is designed to operate on the hidden representations produced by a base model trained on DNA sequences, improving mRNA sequence analysis through a codon-level adapter. Below, we outline the core components in this algorithm.

**Input and hidden representation** The algorithm takes as input hidden states $H \in \mathbb{R}^{\text{batch\_size} \times \text{seq\_len} \times d_{\text{model}}}$, where $H$ is the latent representation generated by a base model trained on nucleotide-level tokenized DNA sequences. These hidden states encapsulate nucleotide-level patterns in the DNA sequence but lack the explicit codon-level representation required for understanding mRNA translation and regulation. CodonMoE restructures these hidden states to focus on codon-level features for better-adapting DNA models for mRNA analysis.

**Codon aggregation and reshaping** mRNA sequences consist of codons, which are triplets of nucleotides fundamental to protein synthesis. The hidden states $H$ are reshaped into groups of three consecutive hidden vectors to form codon-level representations. Specifically, the tensor is reshaped into $[B, S/3, 3d]$, where each codon consists of three concatenated hidden vectors. This step captures interactions between nucleotides within each codon.

**Mixture of Experts (MoE) for codon-level feature learning** At the core of the CodonMoE is a **MoE** mechanism that selects from multiple expert networks to process codon-level representations dynamically. Each codon is processed by num_experts linear sub-networks (experts), where each expert specializes in extracting different semantic aspects of the codon. The outputs of these experts are weighted by a softmax gating mechanism, conditioned on the codon input. This ensures the CodonMoE mechanism is highly adaptable to varying contexts within RNA sequences.

**Codon-level expansion and integration** After extracting codon-level features from the MoE, these features are expanded to match the original sequence length by repeating the codon features three times, once for each nucleotide in the codon. This expanded representation is reshaped back to $[B, S - 1, d]$ and added element-wise to the original hidden states. The result is an enhanced representation that incorporates both nucleotide-level and codon-level information, improving the model's ability to capture local patterns and broader codon interactions.

**Regularization and transformation** To ensure robust learning and prevent overfitting, the algorithm applies a series of regularization and transformation steps:

- **Layer normalization**: Ensures stability during training by normalizing the feature map.

- **GELU activation**: Introduces non-linearity to enhance the model's ability to learn complex relationships between codon sequences and biological function.

- **Dropout**: Prevents overfitting by randomly dropping units during training, particularly useful for high-dimensional biological data.

The final feature map is then flattened and passed through a linear transformation, producing a compact feature vector $Y$ that can be used for downstream tasks, such as mRNA classification or regression.

---

**Algorithm 1** CodonMoE for mRNA Sequence Analysis

---

1: **Input**: Hidden states $H \in \mathbb{R}^{\text{batch\_size} \times \text{seq\_len} \times d_{\text{model}}}$
2: **Output**: Feature vector $Y$
3: **Hyperparameters**: num_experts $\leftarrow 4$, dropout_rate $\leftarrow 0.1$
4: **function** MIXTUREOFEXPERTS($X$)
5:     **for** $i = 1$ to num_experts **do**
6:         $\text{expert}_i \leftarrow \text{Sequential}(\text{Linear}(3d, 3d), \text{GELU}, \text{Linear}(3d, d))$
7:         $\text{outputs}[i] \leftarrow \text{expert}_i(X)$
8:     **end for**
9:     $\text{gate} \leftarrow \text{Softmax}(\text{Linear}(3d, \text{num\_experts})(X))$
10:     **return** $\sum_{i=1}^{\text{num\_experts}} \text{outputs}[i] \odot \text{gate}[:, :, i]$
11: **end function**
12: **function** CODONMOE($H$)
13:
14:     $(B, S, d) \leftarrow \text{shape}(H)$
15:     $Y \leftarrow H[:, : S - 1, :]$
16:     $\text{codons} \leftarrow \text{Reshape}(Y, [B, S//3, 3d])$
17:     $\text{moe} \leftarrow \text{MixtureOfExperts}(\text{codons})$
18:     $\text{expanded} \leftarrow \text{Repeat}(\text{moe}, 3, \dim = 1)$
19:     $\text{expanded} \leftarrow \text{Reshape}(\text{expanded}, [B, S - 1, d])$
20:     $Y \leftarrow Y + \text{expanded}$
21:     $Y \leftarrow \text{Dropout}(\text{GELU}(\text{LayerNorm}(Y)), \text{dropout\_rate})$
22:     $Y \leftarrow \text{Linear}((S - 1)d, d)(\text{Flatten}(Y))$
23:     $Y \leftarrow \text{Dropout}(\text{GELU}(\text{LayerNorm}(Y)), \text{dropout\_rate})$
24:     **return** $\text{Linear}(d, 1)(Y)$
25: **end function**
26: **function** ANALYZE_MRNA(sequence)
27:
28:     $\text{tokens} \leftarrow \text{Tokenize}(\text{sequence})$
29:     $\text{hidden} \leftarrow \text{BaseModel}(\text{tokens})$
30:     **return** $\text{CodonMoE}(\text{hidden})$
31: **end function**

---

A.2 PROOF OF THEOREM 3.3

We aim to show that the CodonMoE, functioning as an adapter to the pretrained DNA backbone $h$, is a universal approximator for any function $f \in \mathcal{F}$, where $\mathcal{F}$ is the class of continuous functions mapping codon sequences to target properties.

Let $x \in \mathcal{X}$ be a sequence partitioned into $n$ codons:

$$x = (c_1, c_2, \ldots, c_n), \quad c_i \in \mathcal{C}.$$

The backbone model $h : \mathcal{X} \to \mathbb{R}^{L \times D}$ with $L = 3n$ generates embeddings:

$$h(x) = [e_1, e_2, \ldots, e_L]^\top \in \mathbb{R}^{L \times D}.$$

Each codon $c_i$ is represented by averaging three nucleotide embeddings:

$$z_i = \frac{e_{3i-2} + e_{3i-1} + e_{3i}}{3} \in \mathbb{R}^D.$$

The CodonMoE applies a Mixture of Experts model to each $z_i$:

$$f_{\text{MoE}}(z_i) = \sum_{k=1}^{K} g_k(z_i) \cdot E_k(z_i),$$

where:

$$g_k(z_i) = \frac{\exp(\phi_k(z_i))}{\sum_{j=1}^{K} \exp(\phi_j(z_i))},$$

with gating functions $\phi_k : \mathbb{R}^D \to \mathbb{R}$, and expert networks $E_k : \mathbb{R}^D \to \mathbb{R}^m$.

By the Universal Approximation Theorem (Hornik et al., 1989), for each $f_k$ and any $\epsilon > 0$, there exists $E_k$ such that:

$$\|E_k(z_i) - f_k(z_i)\| < \frac{\epsilon}{Kn},$$

where $f_k \in C(\mathbb{R}^D, \mathbb{R}^m)$.

Define the overall network function:

$$F(x) = \sum_{i=1}^{n} f_{\text{MoE}}(z_i) = \sum_{i=1}^{n} \sum_{k=1}^{K} g_k(z_i) E_k(z_i).$$

For the target function $f \in \mathcal{F}$, assume:

$$f(x) = \sum_{i=1}^{n} f_i(z_i), \quad f_i \in C(\mathbb{R}^D, \mathbb{R}^m).$$

Then, the approximation error is:

$$\|F(x) - f(x)\| = \left\| \sum_{i=1}^{n} \sum_{k=1}^{K} g_k(z_i) E_k(z_i) - \sum_{i=1}^{n} f_i(z_i) \right\|.$$

Assuming $\sum_{k=1}^{K} g_k(z_i) = 1$ and $g_k(z_i) \geq 0$, we have:

$$\|F(x) - f(x)\| \leq \sum_{i=1}^{n} \sum_{k=1}^{K} g_k(z_i) \|E_k(z_i) - f_i(z_i)\| < \sum_{i=1}^{n} \sum_{k=1}^{K} g_k(z_i) \frac{\epsilon}{Kn} = \frac{\epsilon}{K}.$$

The backbone model $h$ ensures that embeddings $z_i$ capture essential genetic information:

$$h : \mathcal{X} \to \mathbb{R}^{L \times D}, \quad z_i = \mathcal{P}(h(x)),$$

where $\mathcal{P}$ denotes the partitioning into codon embeddings via averaging.

Combining the above, for any $f \in \mathcal{F}$ and $\epsilon > 0$, there exists a CodonMoE network such that:

$$\|F(x) - f(x)\| < \epsilon.$$

Thus, the CodonMoE integrated with the pretrained backbone $h$ satisfies:

$$F = \sum_{i=1}^{n} \sum_{k=1}^{K} g_k(z_i) E_k(z_i) \approx f(x), \quad \forall f \in \mathcal{F}.$$

Therefore, the CodonMoE module, when combined with the pretrained Backbone Model $h$, serves as a universal approximator for any continuous function mapping codon sequences to target properties within the class $\mathcal{F}$.

### A.3 Additional Experimental Details

**Experimental settings.** Table 6 outlines the key components and hyperparameters used for different backbone models, highlighting the settings in regressor types and training parameters such as learning rates and the number of epochs. Specifically, it details the setup for the mRFP expression dataset, using Caduceus and HyenaDNA as primary backbones with variations such as Caduceus-CodonMean and Caduceus-CodonMoE, indicating different CodonMoE variations within the same framework. Specific configurations such as the backbone sequence length, model dimensions, number of layers, and learning rates are listed, with pure backbone models integrating machine learning regressors like MLP and XGBoost. It also outlines settings for the SARS-CoV-2 vaccine degradation dataset with similar backbone models but slightly adjusted parameters, such as a different sequence length for the HyenaDNA models. Both tables showcase the learning rates and epochs where applicable, providing a comprehensive view of how each model is tuned for its respective task.

Table 6: Summary of experimental settings for SARS-CoV-2 vaccine degradation dataset and mRFP expression dataset.

| Backbone | Model | Backbone Name | Regressor | Learning Rate | Epochs |
|---|---|---|---|---|---|
| Caduceus | Caduceus | caduceus-ps_seqlen-1k_d_model-256_n_layer-4_lr-8e-3 | mlp | - | - |
| Caduceus | Caduceus | caduceus-ps_seqlen-1k_d_model-256_n_layer-4_lr-8e-3 | xgboost | - | - |
| Caduceus | Caduceus-CodonMean | caduceus-ps_seqlen-1k_d_model-256_n_layer-4_lr-8e-3 | - | 0.0005 | 100 |
| Caduceus | Caduceus-CodonMoE | caduceus-ps_seqlen-1k_d_model-256_n_layer-4_lr-8e-3 | - | 0.0005 | 100 |
| HyenaDNA | HyenaDNA | hyenadna-small-32k-seqlen | mlp | - | - |
| HyenaDNA | HyenaDNA | hyenadna-small-32k-seqlen | xgboost | - | - |
| HyenaDNA | HyenaDNA-CodonMean | hyenadna-small-32k-seqlen | - | 0.0005 | 100 |
| HyenaDNA | HyenaDNA-CodonMoE | hyenadna-small-32k-seqlen | - | 0.0001(0.001) | 100 |

**Dataset details.** For the mRFP expression dataset, the researchers in the study by Nieuwkoop et al. (2023) constructed low (CAI_L), medium (CAI_M), and high (CAI_H) CAI libraries and expressed them in Escherichia coli DH10B. They quantified mRFP expression using both flow cytometry and microplate reader measurements, normalizing fluorescence to account for variations in cell density. The full-length coding sequence (675 bp) for each variant was determined by Sanger sequencing. They applied quality control criteria to ensure data integrity, excluding samples with low-quality sequencing reads, amino acid mutations, mixed populations, or significant deviations between measurement methods. This curation process resulted in a high-quality dataset that provides a foundation for investigating the determinants of translation efficiency in *E. coli*. We accessed this dataset through the public repository as provided by the original authors and used it as the basis for our machine learning approach to predict protein production levels from mRNA sequence features.

For the SARS-Cov-2 vaccine degradation dataset, this dataset includes mRNA constructs encoding a multi-epitope vaccine (MEV) candidate based on SARS-CoV-2 antigens. The key component of this dataset that we focus on in our experiments is the in-cell mRNA stability via time-course degradation experiments in HEK293T cells. This dataset, as described by Leppek et al. (2022), provides a resource for investigating the relationships between mRNA sequence, structure, stability, and expression efficiency in the context of SARS-CoV-2 vaccine design.

### A.4 Computational Resources

Model training and inference are accomplished on two A100 and two A6000 GPUs.

### A.5 Feature Embedding Visualization

**SARS-CoV-2 vaccine degradation task.** As shown in Figure 3, the UMAP and t-SNE visualizations highlight the CodonMoE model's superior ability to capture fine-grained codon-level patterns and dynamically specialize through its Adaptive Mixture of Experts, resulting in more distinct and diverse clusters compared to the backbone model. CodonMoE's expert system allows for better separation of genetic features, capturing both local codon-specific and broader sequence patterns. This leads to smoother transitions in the continuous target values, as seen in the clearer color gradients in the t-SNE plot, indicating that CodonMoE is able to approximate complex relationships between

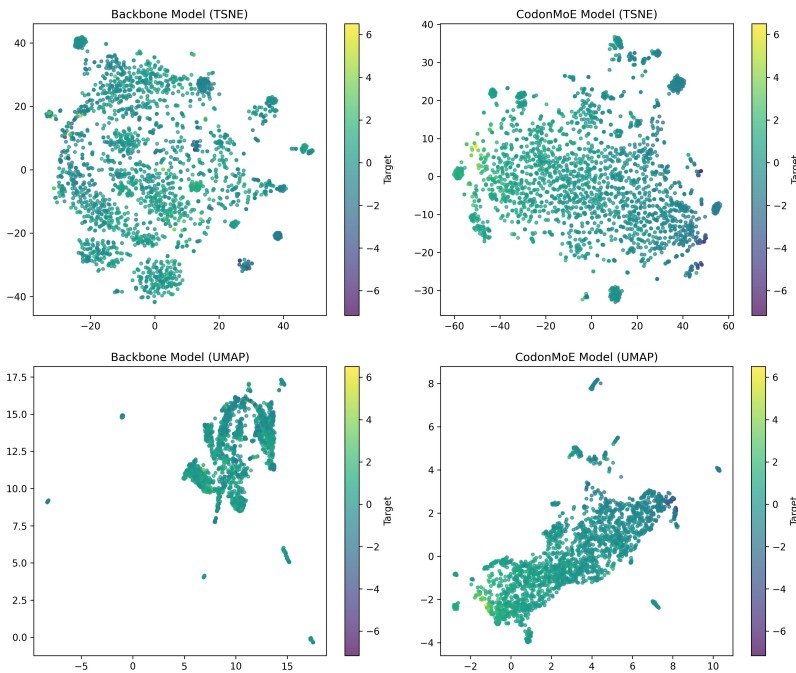

Figure 3: t-SNE and UMAP comparison between features from HyenaDNA model and CodonMoE-enhanced HyenaDNA model on SARS-CoV-2 vaccine degradation dataset.

codon sequences and degradation rates. In contrast, the backbone model's visualizations show more compressed clusters and limited separation, suggesting that it struggles with representing nuanced degradation patterns.

**mRFP expression task.** In Figure 4, the t-SNE and UMAP visualizations highlight the improved performance of the CodonMoE-enhanced HyenaDNA model compared to the backbone model on the mRFP expression dataset. In the t-SNE plot, the backbone model shows tight clusters with limited spread, indicating that it struggles to differentiate between various expression levels, leading to more uniform representations. In contrast, CodonMoE demonstrates broader, more distinct clusters, reflecting its ability to capture finer differences in mRFP expression levels, as seen in the smoother color gradient transitions. Similarly, the UMAP visualization reveals that the backbone model's clusters are tightly packed, suggesting less feature diversity, whereas CodonMoE's clusters are more spread out, indicating richer, more nuanced representations. This enhanced separation and feature diversity in CodonMoE can be attributed to its architecture, which allows it to capture both local codon-level patterns and broader sequence features, resulting in better predictions of continuous targets like mRFP expression levels. Figure 5 shows that the CodonMoE-enhanced GPN-MSA model demonstrates clearer and more distinct clustering. In both t-SNE and UMAP visualizations, the CodonMoE-enhanced backbone features tighter and more defined clusters with a pronounced variation in metric values, suggesting a more effective differentiation.

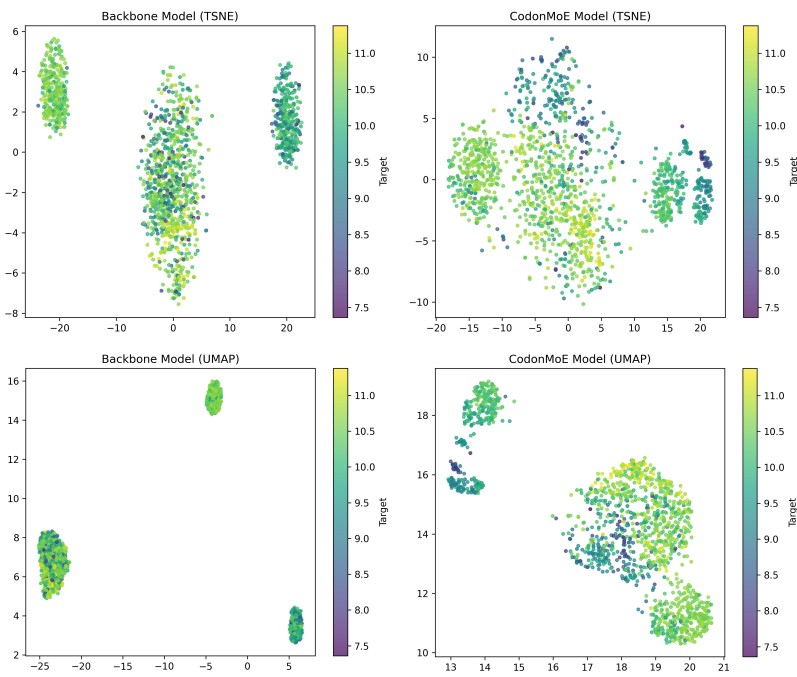

Figure 4: t-SNE and UMAP comparison between features from HyenaDNA model and CodonMoE-enhanced HyenaDNA model on mRFP expression dataset.

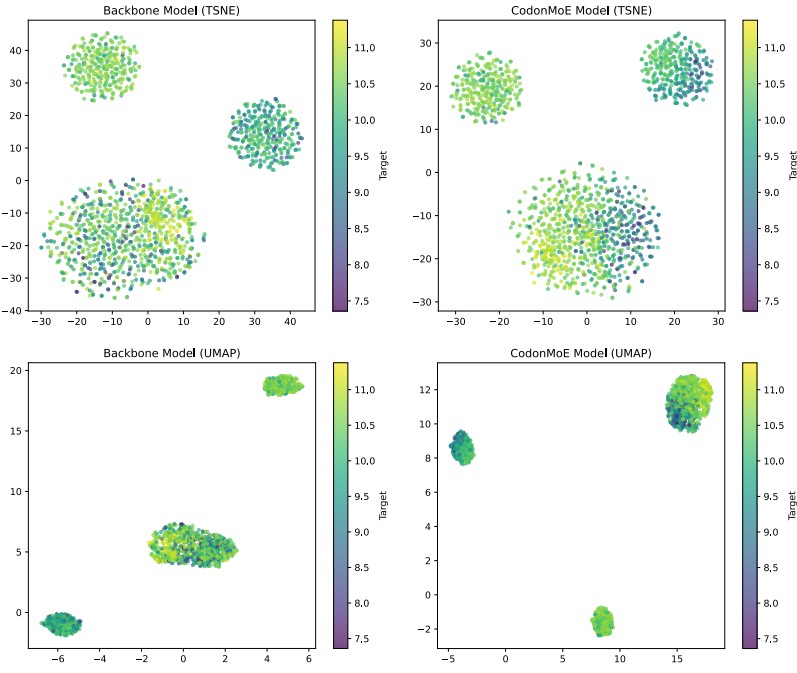

Figure 5: t-SNE and UMAP comparison between features from GPN-MSA model and CodonMoE-enhanced GPN-MSA model on mRFP expression dataset.

## A.6 ADDITIONAL EXPERIMENTS: GPN-SS BACKBONE ENHANCED WITH CODONMOE

The GPN-SS (Genomic Pre-trained Network - Single Sequence) model (Benegas et al., 2023), trained on single-species genomic data, uses convolutional layers to efficiently learn and predict the impacts of genetic variants. This model focuses on analyzing single-species genomes without the confounding effects of cross-species genomic variations, making it valuable for studies targeted at species-specific genomic features. Table 7 shows the comparison of the GPN-SS and GPN-SS-CodonMoE methods in terms of Spearman Rank Correlation metrics for vaccine degradation and mRFP expression, highlighting the universal applicability of our designed module across different backbone architectures and tasks.

Table 7: Evaluation of computational complexity and Spearman's rank correlation metrics based on GPN-SS model.

| Method | Modality | Time Complexity | Model Parameters | Vaccine Degradation | mRFP expression |
|--------|----------|-----------------|------------------|---------------------|-----------------|
| GPN-SS | DNA | linear | >50M | 0.60 | 0.56 |
| GPN-SS-CodonMoE | DNA | linear | >50M | 0.74 | 0.82 |

## A.7 UPDATED MAIN TABLE WITH DETAILED PARAMETERS AND ENHANCED MODELS

In the updated main table (Table 8), we provide a comprehensive evaluation of computational complexity and Spearman's rank correlation metrics across various RNA models and CodonMoE-enhanced DNA backbones. This update primarily focuses on comparing detailed backbone parameters, introducing a new framework, and detailing the performance improvements achieved through our proposed modifications.

A significant addition to our evaluation is the introduction of the HyenaDNA-CodonMoE$_{TextCNN}$ framework. In this variant, the traditional MLPs within the CodonMoE module are replaced with TextCNN architectures. This substitution leverages the strengths of convolutional neural networks in capturing local patterns and hierarchical features within genomic data. By integrating TextCNN in place of MLPs, the CodonMoE module becomes more adept at handling the sequential and spatial dependencies inherent in DNA sequences. This architectural enhancement not only improves the model's ability to extract meaningful representations from the data but also maintains a balance between computational efficiency and performance.

The introduction of the HyenaDNA-CodonMoE$_{TextCNN}$ variant further elevates performance by effectively replacing the MLP with the TextCNN, resulting in more robust and accurate predictions. This variant achieves performance levels that rival the top-performing RNA models while maintaining lower computational complexity. The enhanced ability to capture intricate patterns within the genomic data without a significant increase in model parameters underscores the effectiveness of the CodonMoE module in optimizing both performance and efficiency.

Additionally, we provide detailed parameters for the primary frameworks under comparison, including HyenaDNA-CodonMoE, HyenaDNA-CodonMoE$_{CNN}$, and HyenaDNA, evaluated for both performance and parameter efficiency, alongside the top-performing RNA-specific model Codon-BERT.

Overall, the updated evaluations confirm that the integration of the CodonMoE module is a robust strategy for enhancing model performance across different DNA backbones. The introduction of the HyenaDNA-CodonMoE$_{TextCNN}$ framework, in particular, sets a new standard by balancing high performance with computational efficiency. These advancements demonstrate the potential of our proposed modifications in developing more scalable and effective language models for genomic research, offering improved tools for understanding and manipulating genetic information with reduced computational overhead.

Table 8: Evaluation of computational complexity and Spearman's rank correlation metrics across RNA and DNA language models: CodonMoE suffix indicates models enhanced with our proposed CodonMoE module. Each data set is split into training, validation, and testing with a 0.7, 0.15, and 0.15 ratio, using the same split set as in the CodonBERT (Li et al., 2024). The metric is Spearman's rank Correlation.

| Method | Modality | Time Complexity | Model Parameters | Vaccine Degradation | mRFP Expression |
|---|---|---|---|---|---|
| RNA Models | | | | | |
| CodonBERT | RNA | quadratic | 81.7M | **0.77** | **0.85** |
| DNA Models | | | | | |
| GPN-MSA | DNA | quadratic | 85.7M | 0.55 | 0.33 |
| GPN-MSA-CodonMoE | DNA | quadratic | 161.9M | 0.77 | 0.79 |
| GPN-MSA-CodonMoE$_{TextCNN}$ | DNA | quadratic | 115.0M | 0.82 | 0.81 |
| HyenaDNA | DNA | subquadratic | 4.1M | 0.69 | 0.44 |
| HyenaDNA-CodonMoE | DNA | subquadratic | 12.7M | 0.81 | 0.84 |
| HyenaDNA-CodonMoE$_{TextCNN}$ | DNA | subquadratic | 7.5M | **0.84** | **0.85** |

## A.8 ADDITIONAL EXPERIMENTS OF COMPARATIVE ANALYSIS OF CODONOPERATOR VARIANTS: CODONMOE$_{TEXTCNN}$

As shown in Table 9 and Table 10, building upon CodonMoE, we introduced an additional variant, CodonMoE$_{TextCNN}$, which replaces the MLP layers within CodonMoE with a Text Convolutional Neural Network (TextCNN). The TextCNN configuration was adapted from RNAFM$_{TextCNN}$ and RNABERT$_{TextCNN}$, aiming to better capture local sequence patterns and enhance the model's ability to discern complex codon-level dependencies.

By replacing the MLP layers with TextCNN, CodonMoE$_{TextCNN}$ leverages convolutional operations to effectively model local sequence patterns, a strategy adapted from RNAFM$_{TextCNN}$ and RNABERT$_{TextCNN}$. This architectural modification enhances the model's ability to detect and utilize fine-grained codon interactions, thereby improving overall predictive performance.

Table 9: CodonOperator variant comparison (including CodonMoE$_{TextCNN}$) on mRFP expression dataset.

| | GPN-MSA | HyenaDNA |
|---|---|---|
| CodonMean | 0.740 | 0.765 |
| CodonMoE | 0.790 | 0.837 |
| CodonMoE$_{TextCNN}$ | **0.808** | **0.851** |

Table 10: CodonOperator variant comparison (including CodonMoE$_{TextCNN}$) on SARS-CoV-2 vaccine degradation dataset.

| | GPN-MSA | HyenaDNA |
|---|---|---|
| CodonMean | 0.729 | 0.789 |
| CodonMoE | 0.770 | 0.812 |
| CodonMoE$_{TextCNN}$ | **0.823** | **0.844** |

## A.9 ADDITIONAL EXPERIMENTS ON EVALUATION OF DNA PRETRAINED MODEL FEATURE EFFECTIVENESS

In this section, we explore the ability of DNA-pretrained backbones, specifically Caduceus and HyenaDNA, to effectively generalize to mRNA-related tasks using a TextCNN framework. The tasks evaluated include predictions on the SARS-CoV-2 vaccine degradation dataset and the mRFP expression dataset. The scatter plots in Figure 6 provide a visual representation of the alignment between actual and predicted values, with a trendline indicating overall correlation.

The SARS-CoV-2 vaccine degradation dataset serves as a proxy for evaluating the potential of DNA-pretrained features to capture complex biological dependencies related to RNA sequence stability and degradation. Both models demonstrate a clear trend of alignment between actual and predicted values, reflecting the potential of DNA-derived features to transfer effectively to mRNA stability prediction. Despite the inherent challenges of modeling degradation, as indicated by a wider spread in predictions, the performance reflects the potential of pretrained DNA models to generalize beyond their training domain to tasks with overlapping biological mechanisms, such as RNA stability. The effectiveness of these features suggests that key structural and sequence-specific attributes learned from DNA datasets are applicable to mRNA-related degradation tasks.

The mRFP expression dataset focuses on the predictability of gene expression levels based on underlying sequence features. Both models achieve a closer alignment of predicted values to the actual values compared to the degradation dataset. This suggests that the DNA-pretrained features can be potentially effective at tasks involving expression prediction, where sequence features such as promoter regions, codon optimization, and untranslated regions are critical. The high clustering around the trendline demonstrates that these DNA backbones successfully capture sequence motifs and structural patterns that are transferable to mRNA-related tasks. This finding aligns with the hypothesis that DNA and RNA share significant overlapping biological motifs, enabling effective transfer learning.

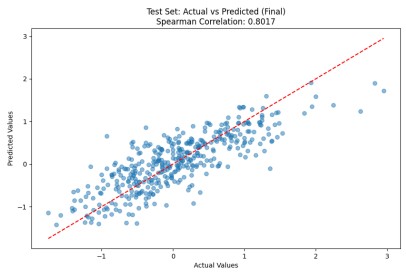

(a) Caduceus pretrained model feature effectiveness on SARS-CoV-2 vaccine degradation dataset.

(b) Caduceus pretrained model feature effectiveness on mRFP expression dataset.

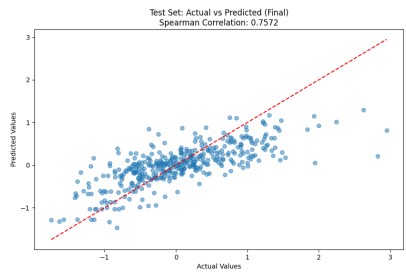

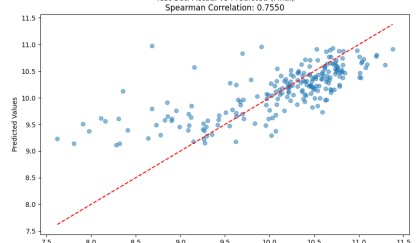

(c) HyenaDNA pretrained model feature effectiveness on SARS-CoV-2 vaccine degradation dataset.

(d) HyenaDNA pretrained model feature effectiveness on mRFP expression dataset.

Figure 6: Evaluation of DNA pretrained model feature effectiveness on mRFP expression and SARS-CoV-2 vaccine degradation dataset using TextCNN.

## A.10    UPDATED ABLATION STUDIES (INCLUDING TEXTCNN)

In this section, we present the updated ablation studies (Table 12 and Table 11) that incorporate the TextCNN architecture alongside the previously evaluated MLP and XGBoost models. These studies assess the effectiveness of features extracted from DNA pretrained models—namely GPN-MSA, HyenaDNA, and Caduceus—on two proposed datasets.

Table 11 evaluates the performance of MLP, XGBoost, and TextCNN on the mRFP expression dataset using features extracted from the DNA pretrained models. The results indicate that TextCNN significantly outperforms both MLP and XGBoost across all three models, achieving the highest Spearman's rank correlation scores. Specifically, TextCNN exhibits a marked improvement in correlation metrics, suggesting its superior ability to capture and leverage the intricate patterns within the feature representations derived from the DNA models.

Similarly, Table 12 presents the evaluation on the SARS-CoV-2 vaccine degradation dataset. While XGBoost remains the top performer for GPN-MSA, TextCNN surpasses XGBoost for HyenaDNA and Caduceus, achieving the highest correlation scores. This indicates that TextCNN not only excels in tasks where XGBoost previously dominated but also provides consistent performance improvements across different DNA backbones. The ability of TextCNN to handle sequential and spatial dependencies more effectively than traditional regression models like MLP and XGBoost highlights its potential as a superior architecture for downstream genomic tasks.

The updated ablation studies conclusively demonstrate that the inclusion of TextCNN within the CodonMoE module significantly enhances the performance of DNA pretrained models on relevant genomic tasks. These findings highlight the importance of architectural choices in model design and support the efficacy of our proposed CodonMoE enhancements in achieving a balance between performance and computational efficiency.

Table 11: Evaluation of DNA pretrained model feature effectiveness on mRFP expression dataset using MLP, XGBoost and TextCNN.

|  | GPN-MSA | HyenaDNA | Caduceus |
| --- | --- | --- | --- |
| MLP | 0.330 | 0.439 | 0.490 |
| XGBoost | 0.479 | 0.512 | 0.476 |
| TextCNN | **0.758** | **0.755** | **0.785** |

Table 12: Evaluation of DNA pretrained model feature effectiveness on SARS-CoV-2 vaccine degradation dataset using MLP, XGBoost and TextCNN.

|  | GPN-MSA | HyenaDNA | Caduceus |
| --- | --- | --- | --- |
| MLP | 0.572 | 0.695 | 0.560 |
| XGBoost | **0.750** | 0.711 | 0.737 |
| TextCNN | 0.717 | **0.757** | **0.801** |

## A.11 ADDITIONAL ABLATION STUDIES

To further evaluate the effectiveness of the CodonMoE architecture, we conducted additional experiments comparing its performance with a dense baseline model. The results are summarized in Table 13. The dense baseline replaces the CodonMoE module with standard dense layers while maintaining an equivalent number of trainable parameters and identical training hyperparameters, ensuring a controlled setup for fair ablation studies. This approach isolates the contribution of the CodonMoE architecture to the overall performance.

The consistent performance gains across both datasets indicate that CodonMoE's specialized design provides superior modeling capabilities compared to standard dense layers under matched parameter constraints. This reinforces the potential of CodonMoE as a plug-and-play module for adapting DNA-based models to mRNA tasks, offering both computational efficiency and improved predictive performance.

Table 13: Performance comparison between the standard dense baseline and HyenaDNA-CodonMoE$_{\text{TextCNN}}$ (equivalent parameters) on SARS-CoV-2 vaccine degradation dataset and RFP expression dataset.

| Model | Vaccine Degradation | mRFP Expression |
|---|---|---|
| HyenaDNA-Densebaseline$_{\text{TextCNN}}$ | 0.80 | 0.82 |
| HyenaDNA-CodonMoE$_{\text{TextCNN}}$ | **0.84** | **0.85** |

## A.12 ADDITIONAL INTRODUCTION OF DNA BACKBONES

### A.12.1 RNABERT

RNABERT is a nucleotide-based RNA large language model trained on non-coding RNAs (ncR-NAs) to provide effective embeddings of RNA bases. It integrates context-sensitive nucleotide information with secondary structural features to enhance its understanding of RNA functionality. Trained on 76,237 non-coding RNA sequences from RNAcentral using masked language modeling and structural alignment learning, RNABERT excels in capturing both nucleotide-level interactions and higher-order structural similarities that underpin RNA functionality. The architecture of RNABERT, comprising 6 Transformer layers with a hidden dimension of 120.

### A.12.2 RNA-FM

RNA-FM is a nucleotide-based foundational RNA language model specifically designed for large-scale RNA structure and function prediction. RNA-FM employs a 12-layer bidirectional Transformer encoder to capture intricate long-range interactions and evolutionary signals within RNA sequences. Trained on 23 million unannotated ncRNA sequences from RNAcentral using self-supervised learning, RNA-FM generates highly expressive embeddings that represent both structural and functional characteristics. Despite its larger architecture, RNA-FM demonstrates high efficiency, offering robust generalization across diverse RNA datasets while requiring less fine-tuning for new tasks. Its flexibility and precision make RNA-FM a cornerstone model for advancing RNA research across multiple domains.

### A.12.3 CODONBERT

CodonBERT is a codon-based RNA language model built on the BERT architecture, featuring a 12-layer bidirectional Transformer encoder with 12 self-attention heads per layer and a hidden dimension of 768 at each position. It is pre-trained on 10 million mRNA coding sequences (CDS) sourced from NCBI, covering mammals, bacteria, and human viruses across 13 evolutionary categories. Input sequences are split into codons (triplets of nucleotides) and encoded through a combination of codon embeddings, positional embeddings, and segment embeddings, resulting in context-aware codon representations for downstream tasks. In addition to the Masked Language Modeling (MLM) task, CodonBERT incorporates Homologous Sequence Prediction (HSP), where pairs of mRNA sequences are classified to determine their evolutionary relationships, aiding in the learning of sequence homology. The sequences are preprocessed to ensure lengths are multiples of three, beginning with the start codon (AUG) and ending with stop codons (UAA, UAG, or UGA). Compared to RNABERT and RNA-FM, which focus on nucleotide-based embeddings and non-coding RNA, CodonBERT leverages codon-level inputs, providing a deeper understanding of translation-related features and evolutionary information, making it particularly effective for tasks like mRNA optimization and protein expression prediction.

### A.12.4 GPN-MSA

GPN-MSA is a DNA language model optimized for genome-wide variant effect prediction, utilizing a multiple-sequence alignment (MSA) of 100 vertebrate species. These alignment blocks are then stitched together using the multiz utility maf2fasta, ensuring that any columns with gaps in the human reference are removed, and excluding the 10 primate species closest to humans to avoid bias from excessive similarity. Additionally, associated conservation scores from phastCons and phyloP, which provide important information about evolutionary conservation across species, are downloaded and integrated into the training data.

The GPN-MSA model architecture leverages masked language modeling techniques, using a 128-bp multiple-sequence alignment (MSA) window. In this setup, 15% of the positions within the human reference sequence are masked randomly during training, and the model learns to predict these nucleotides based on the contextual information provided by both the positions and species represented in the MSA. The sequence of MSA columns is processed through a Transformer neural network named RoFormer (Su et al., 2024), which results in a high-dimensional contextual embedding for each position, and a final layer outputs the probabilities for four nucleotides at each masked position.

To optimize the learning process, the model downweights repetitive elements and upweights conserved elements, ensuring that incorrect predictions in neutral regions are penalized less severely. A smoothed version of phastCons, referred to as phastConsM, is used to emphasize highly conserved regions and those immediately adjacent to them. As part of data augmentation in non-conserved regions, the reference nucleotide is replaced by a random nucleotide with a certain probability, guiding the model to assign more neutral scores in these less conserved areas. This strategic integration of evolutionary conservation and species diversity, along with sophisticated neural modeling techniques, allows GPN-MSA to effectively learn from a rich and complex set of genomic data, making it a powerful tool for predicting variant effects across the genome.

### A.12.5 HYENADNA

HyenaDNA is a genomic foundation model that addresses the challenges of long-range dependencies and single-nucleotide resolution in DNA sequence analysis. Unlike traditional Transformer-based approaches constrained by the quadratic scaling of attention mechanisms, HyenaDNA employs the Hyena operator, which enables ultralong context lengths of up to 1 million tokens. This represents a 500x improvement in context length over previous dense-attention genomic models. Pretrained on the human reference genome using next-nucleotide prediction, HyenaDNA excels in capturing both the intricate long-range interactions within genomic sequences and the subtle single-nucleotide variations that drive biological functions. Its architecture is highly efficient, scaling sub-quadratically in sequence length and training up to 160x faster than Transformers for similar tasks. Despite using significantly fewer parameters and less pretraining data, HyenaDNA achieves state-of-the-art performance across 20+ genomic benchmarks, including enhancer identification and chromatin profile prediction. Moreover, its innovative use of soft prompting and in-context learning allows for rapid adaptation to new genomic tasks without fine-tuning model weights, showcasing its flexibility and broad utility in genomic research.

### A.12.6 CADUCEUS

Caduceus is a DNA language model that combines novel architectural innovations to address critical challenges in genomic sequence modeling, including long-range dependencies, bi-directionality, and reverse complement (RC) equivariance. Unlike traditional genomic models, Caduceus leverages the MambaDNA block, a powerful extension of the Mamba module, to process sequences bi-directionally while incorporating RC-equivariant processing as an inductive bias. This ensures that predictions remain invariant under strand reversal, a critical requirement for accurate DNA sequence modeling.

Pretrained on the human reference genome with a masked language modeling (MLM) objective, Caduceus is specifically designed to handle sequences extending to hundreds of thousands of nucleotides, surpassing the limitations of unidirectional models or those reliant on quadratic scaling attention mechanisms. Its RC-equivariant embeddings and prediction heads enhance its ability to capture the symmetry of DNA, making it particularly effective in tasks involving regulatory annotations, enhancer prediction, and variant effect analysis.

The model achieves exceptional performance across a broad range of genomic tasks, including variant effect prediction and enhancer classification, often outperforming significantly larger models such as Nucleotide Transformer v2 (Dalla-Torre et al., 2023) and other Transformer-based architectures.

## A.13 UPDATED DATASET INTRODUCTION

The Tc-riboswitch dataset (Groher et al., 2018) was developed to optimize the dynamic range (DR) and basal expression (BE) of tetracycline (Tc)-responsive synthetic riboswitches. These constructs consist of tandem Tc aptamers inserted into the $5'$ untranslated region (UTR) of a GFP reporter gene, regulating expression in response to Tc ligand binding.

Using *Saccharomyces cerevisiae* RS453 as the host, GFP fluorescence was quantified with and without Tc induction via flow cytometry. Through machine learning-guided optimization, including random forest classifiers and convolutional neural networks, sequence and structural features influencing DR and BE were systematically explored. The curated dataset includes constructs with optimized biophysical properties, providing a foundation for understanding riboswitch function and advancing ML-driven design frameworks.

## A.14 UPDATED MAIN TABLE WITH A NEW DATASET

Additional experiments were conducted on the Tc-Riboswitches dataset, as presented in Table 14. The selected datasets—mRFP expression, SARS-CoV-2 vaccine degradation, and Tc-Riboswitches—were chosen for their relevance and diversity in capturing critical aspects of mRNA functionality, such as protein expression levels, structural stability, and regulatory mechanisms. Together, these tasks provide a robust framework for evaluating CodonMoE's capability to address diverse challenges associated with mRNA analysis.

Table 14: Evaluation of computational complexity and Spearman's rank correlation metrics across RNA and DNA language models on Tc-riboswitches dataset. Each data set is split into training, validation, and testing with a 0.7, 0.15, and 0.15 ratio, using the same split set as in the CodonBERT (Li et al., 2024). The metric is Spearman's rank Correlation.

| Method | Modality | Time Complexity | Model Parameters | Tc-Riboswitch |
|---|---|---|---|---|
| | | RNA Models | | |
| RNABERT$_{+TextCNN}$ | RNA | quadratic | 0.48M | 0.47 |
| RNA-FM$_{+TextCNN}$ | RNA | quadratic | 100M | **0.58** |
| CodonBERT | RNA | quadratic | 81.7M | 0.56 |
| | | DNA Models | | |
| HyenaDNA | DNA | subquadratic | 4.1M | 0.40 |
| HyenaDNA-CodonMoE$_{TextCNN}$ | DNA | subquadratic | 7.5M | 0.56 |

