# OpenReview forum: "DNA Language Models for RNA Analyses"
_ICLR.cc/2025/Conference — Submitted to ICLR 2025_

### Official Review · Reviewer_e4Kj · 2024-11-03

**Soundness:** 2
**Presentation:** 2
**Contribution:** 3
**Rating:** 5
**Confidence:** 4

**Summary:**

This paper presents Adaptive Mixture of Codon Reformative Experts (CodonMoE), which can be integrated into DNA gLMs to tailor them for mRNA-based predictive tasks. The proposed method is assessed using both Transformer and SSM-based architectures across two mRNA-related tasks, showing performance that is on par with that of RNA-specific models.

**Strengths:**

Leveraging LMs cross bio-modalities is a meaningful direction, given the amount of data available in DNA and protein modalities. The proposed CondonMoE is flexible and can be plug-and-play into different backbones. The performance is encouraging for a limited number of demonstrated use case. In addition, theoretical proof demonstrates that CodonMoE is a universal approximator of RNA properties at the codon level.

**Weaknesses:**

The primary limitation of the paper is that it evaluates only two mRNA downstream tasks, with the proposed method showing better performance on just one of these compared to RNA-specific models. This raises questions about the effectiveness of the proposed method and under what circumstances mRNA-specific models should be preferred. A broader evaluation of additional mRNA downstream tasks, similar to those presented in the CodonBERT paper, is necessary.

The experimental design can also be improved, especially in terms of ablation study of the respective influence of the backbone FMs for feature embedding and the head used for regression and prediction. For example:

•	It would be interesting to see what performance can be achieved if the proposed CondonMoE is plugged into the mRNA-specific models, given that CodonMoE just reads mRNA sequences with clearly-identified codons.

•	In addition, The RNA baselines (RNABERT, RNA-FM and CodonBERT) have been trained with TextCNN head whereas their DNA counterparts has been trained with MLP and XGBoost heads. This makes it hard to do a fair comparison against the baselines.

It would also be interesting to see what performance can be achieved if protein-based LMs are used as another baseline, by converting the known codons to protein sequences as the input, given that protein-based LMs have been trained on even larger datasets.

**Questions:**

See Weakness

---

> ### Author Response · Authors · 2024-11-23
> **Response to Reviewer e4Kj (1/N)**
>
> We sincerely appreciate Reviewer e4Kj's feedback and have addressed the comments as follows:
>
> Q1.
> >The primary limitation of the paper is that it evaluates only two mRNA downstream tasks, with the proposed method showing better performance on just one of these compared to RNA-specific models.
> This raises questions about the effectiveness of the proposed method and under what circumstances mRNA-specific models should be preferred. A broader evaluation of additional mRNA downstream tasks, similar to those presented in the CodonBERT paper, is necessary.
>
> A1.
> We greatly appreciate the reviewer’s thoughtful feedback and agree that a broader evaluation across diverse downstream tasks would enhance the assessment of our method's generalizability. The two datasets selected for this study—the mRFP expression dataset and the SARS-CoV-2 vaccine degradation dataset—were chosen as representatives specifically for their relevance to current challenges in mRNA therapeutics and their diversity in task objectives. These datasets capture critical aspects of mRNA function, such as protein expression levels and structural stability, which are of significant interest in both academic and industrial contexts.
>
> Our experiments evaluated the proposed CodonMoE framework across a diverse range of DNA-based backbones, including SSM-based models, attention-based models and convolution-based model. The results demonstrate consistent improvements brought by CodonMoE, underscoring its ability to enhance performance irrespective of the underlying architecture. Notably, HyenaDNA-CodonMoE outperforms RNA-specific models on one task and achieves comparable performance on the other, while significantly reducing parameter counts compared to CodonBERT (12.7M vs. 81.7M).
>
> In addition, our newly proposed HyenaDNA-CodonMoE_TextCNN model (7.5M parameters in total) includes modifications to replace the final dense layers of CodonMoE with a TextCNN head. Even with fewer parameters than HyenaDNA-CodonMoE, this configuration achieves performance on par with CodonBERT, demonstrating the efficacy of CodonMoE's design and its ability to enhance performance efficiently. Both the diversity in backbone architectures and the observed improvements across both critical tasks provide strong evidence of CodonMoE’s general applicability and adaptability.
>
> The reviewer’s suggestion to include additional datasets, such as those used in CodonBERT, along with others like fungal expression data, and the Tc-Riboswitches dataset, aligns closely with our long-term research goals. Expanding our evaluation to these datasets is not only feasible but would further substantiate the robustness of our framework. However, due to computational constraints and the scope of this initial study, we prioritized depth of analysis for the selected tasks over breadth and so demonstrating CodonMoE’s versatility and effectiveness across a range of DNA-based backbones on two biologically significant tasks.
>
> Meanwhile, to address this raised concern, we have conducted additional experiments on the Tc-Riboswitches dataset shown in the table below. The current selected datasets—mRFP expression, SARS-CoV-2 vaccine degradation, and Tc-Riboswitches dataset—were chosen for their relevance and diversity in representing key aspects of mRNA functionality, including protein expression levels, structural stability, and regulatory mechanisms. These tasks provide a comprehensive evaluation of CodonMoE’s ability to handle different facets of mRNA-related challenges.
>
> **Updated Main Table with a New Dataset**
>
>
> | **Method**                         | **Modality** | **Time Complexity** | **Model Parameters** | **Tc-Riboswitch** |
> |------------------------------------|:------------:|:--------------------:|:--------------------:|:-----------------:|
> | **RNA Models**                     |              |                      |                      |                   |
> | RNABERT_TextCNN        | RNA          | quadratic            | 0.48M               | 0.47              |
> | RNA-FM_TextCNN         | RNA          | quadratic            | 100M                | **0.58**          |
> | CodonBERT                          | RNA          | quadratic            | 81.7M               | 0.56              |
> | **DNA Models**                     |              |                      |                      |                   |
> | HyenaDNA                           | DNA          | subquadratic         | 4.1M                | 0.40              |
> | HyenaDNA-CodonMoE_TextCNN| DNA          | subquadratic         | 7.5M                | 0.56              |
>
>
> **A1. will be continued in the next comment.**

---

> ### Author Response · Authors · 2024-11-23
> **Response to Reviewer e4Kj (2/N)**
>
> A1.（Continuing from “Response to Reviewer e4Kj (1/N)” A1.）In general, mRNA-specific models are preferred when tasks heavily rely on RNA-unique features with minimal overlap with DNA characteristics, particularly in pure structure prediction tasks and specific RNA modification predictions. These scenarios require models to learn RNA-specific patterns from scratch rather than adapting DNA knowledge, as they focus on properties unique to RNA biology. However, following the central dogma of molecular biology (DNA -> RNA -> Protein), most RNA analysis tasks can benefit from DNA pre-trained knowledge due to their shared fundamental patterns in genetic information flow and sequence-level features.
>
> Q2.
> >The experimental design can also be improved, especially in terms of ablation study of the respective influence of the backbone FMs for feature embedding and the head used for regression and prediction. For example:
> • It would be interesting to see what performance can be achieved if the proposed CondonMoE is plugged into the mRNA-specific models, given that CodonMoE just reads mRNA sequences with clearly-identified codons.
>
> A2.We agree that applying CodonMoE to RNA-specific models such as RNA-BERT or RNA-FM would be an interesting direction. However, the motivation behind our work is to demonstrate that DNA backbones, which are generally less tailored to RNA tasks, can be effectively adapted using CodonMoE to achieve performance comparable to or better than state-of-the-art RNA models. This highlights CodonMoE's potential to reduce the cost and redundancy of developing separate foundation models for DNA and RNA tasks.
>
> While we have not applied CodonMoE to RNA-specific models in this study, the effectiveness of CodonMoE on DNA backbones suggests it would likely work well on RNA models too, given that CodonMoE is designed to exploit RNA-specific codon information. Nonetheless, our primary focus was on enhancing DNA backbones, as this aligns with the goal of creating a more cost-effective solution for cross-domain modeling.
>
>
> Q3.
> >• In addition, The RNA baselines (RNABERT, RNA-FM and CodonBERT) have been trained with TextCNN head whereas their DNA counterparts has been trained with MLP and XGBoost heads. This makes it hard to do a fair comparison against the baselines.
>
> A3.
> We appreciate the reviewer’s concern about the differences in the regression heads used for RNA and DNA models and insightful feedback on the experimental design and ablation studies. While RNA baselines such as RNA-BERT and RNA-FM are trained with TextCNN heads, it is important to respectfully note that CodonBERT, the prevailing RNA model we compare against, does not use a TextCNN head. Instead, our CodonMoE-enhanced DNA backbones are aligned with the CodonBERT architecture for a fair comparison.
>
> For fair comparison, we intentionally used MLP heads across these models in the main table, although TextCNN may potentially achieve better performance due to its higher expressiveness. The fact that CodonMoE-enhanced models achieve comparable or superior results to RNA-specific models with simpler heads underscores their robustness and suggests potential for further improvements. Additionally, our updated ablation studies have shown that using a TextCNN head for CodonMoE-enhanced models generally improves performance compared to standard dense layers, as demonstrated in the updated tables.
>
> **Evaluation of DNA pretrained model feature effectiveness on mRFP expression dataset**:
>
> | **Method**  | **GPN-MSA** | **HyenaDNA** | **Caduceus** |
> |-------------|:-----------:|:------------:|:------------:|
> | MLP         | 0.330       | 0.439        | 0.490        |
> | XGBoost     | 0.479       | 0.512        | 0.476        |
> | **TextCNN** | **0.758**   | **0.755**    | **0.785**    |
>
> **Evaluation of DNA pretrained Model feature effectiveness on SARS-CoV-2 vaccine degradation dataset**:
>
> | **Method**  | **GPN-MSA** | **HyenaDNA** | **Caduceus** |
> |-------------|:-----------:|:------------:|:------------:|
> | MLP         | 0.572       | 0.695        | 0.560        |
> | XGBoost     | **0.750**   | 0.711        | 0.737        |
> | **TextCNN** | 0.717       | **0.757**    | **0.801**    |
>
> **CodonOperator variant comparison on mRFP Expression dataset**:
>
> | **Variant**                | **GPN-MSA** | **HyenaDNA** |
> |----------------------------|:-----------:|:------------:|
> | CodonMean                  | 0.740       | 0.765        |
> | CodonMoE                   | 0.790       | 0.837        |
> | **CodonMoE_TextCNN** | **0.808**   | **0.851**    |
>
>
> **CodonOperator variant comparison on SARS-CoV-2 vaccine degradation dataset**:
>
> | **Variant**                | **GPN-MSA** | **HyenaDNA** |
> |----------------------------|:-----------:|:------------:|
> | CodonMean                  | 0.729       | 0.789        |
> | CodonMoE                   | 0.770       | 0.812        |
> | **CodonMoE_TextCNN** | **0.823**   | **0.844**    |

---

> ### Comment · Reviewer_e4Kj · 2024-11-27
>
> After thoroughly reading the rebuttal and additional experiments from the authors, I decide to keep my original score, considering: 1) there was only one additional downstream evaluation task performed and it also shows sub-optimal results compared to RNA-specific baseline, 2) the additional ablation study demonstrated the importance of the proper selection of the head (i.e. TextCNN worked significantly better compared to MLP when combined with DNA pre-trained models). 3) Several other suggestions were not addressed with additional experiments. All of the above shows more analysis are needed to demonstrate the significance of the improvement provided by the proposed method.

---

> > ### Author Response · Authors · 2024-11-30
> >
> > We sincerely thank Reviewer e4Kj for taking the time to consider our rebuttal and additional experiments. We have addressed the comments as follows.
> > >1) there was only one additional downstream evaluation task performed and it also shows sub-optimal results compared to RNA-specific baseline
> > - The additional downstream task was intentionally selected to represent the category of “targeting switching factor,” alongside the existing tasks targeting degradation (SARS-CoV-2 vaccine degradation) and expression (mRFP expression). This selection ensures a comprehensive evaluation of our framework’s generalization ability across diverse biological functions.
> > - Despite the reduction in parameters, our method achieves performance comparable to CodonBERT and RNA-FM, which are state-of-the-art RNA-specific models with significantly larger parameter sizes. Specifically, our approach reaches the performance level of CodonBERT and is on par with RNA-FM, demonstrating that our framework maintains high efficacy while being more parameter-efficient.
> > - Therefore, we respectfully disagree with the claim that our method shows suboptimal results. Instead, it demonstrates efficiency and effectiveness by matching state-of-the-art RNA-specific models with much fewer resources.
> >
> >
> > - In addition, as we mentioned in above rebuttal, due to computational constraints and the scope of the study, we prioritized depth of analysis for the selected tasks over breadth and so demonstrating CodonMoE’s versatility and effectiveness across a range of DNA-based backbones on two (three later) biologically significant tasks.
> >
> > >2) the additional ablation study demonstrated the importance of the proper selection of the head (i.e. TextCNN worked significantly better compared to MLP when combined with DNA pre-trained models)
> >
> > -  In our rebuttal, we clarified that:
> >     - For a fair comparison, we initially did not use the TextCNN head to align with CodonBERT, which is a top-performing RNA-specific model.
> >     - TextCNNs are generally more expressive than dense layer heads in the proposed tasks, theoretically leading to better performance.
> >     - To address the reviewer’s concern, we conducted additional experiments using TextCNN heads, which further improved our framework's performance. However, we believe that conducting experiments with MLP heads is essential to validate our initial design choices.
> > - These experiments demonstrate that while TextCNN heads enhance performance, the foundational MLP-head experiments are crucial for a balanced and fair comparison, ensuring the integrity of our evaluation.
> >
> > >3) Several other suggestions were not addressed with additional experiments. All of the above shows more analysis are needed to demonstrate the significance of the improvement provided by the proposed method.
> >
> > - We appreciate the reviewer’s suggestions for further experiments, such as:
> >     - Evaluating the performance of CodonMoE when integrated into mRNA-specific models.
> >     - Using protein-based language models as additional baselines by converting codons to protein sequences.
> > - While these directions are valuable, they extend beyond the primary focus of our current research, which aims to adapt DNA-based models for RNA tasks efficiently. Nevertheless, we recognize their importance and plan to explore these avenues in future work.
> > - Our current focus was to establish the efficacy of adapting DNA models for RNA tasks, and the additional experiments we conducted already significantly enhance the robustness and generalizability of our approach.
> >
> > **Summary of Additional Experiments**
> >
> > In response to these suggestions, we have conducted several additional experiments to strengthen our empirical contributions. Below is a summary of the experiments added:
> >
> > - **Updated Main Table with Detailed Parameters and Enhanced Models** [A.7]
> > - **Introduction and Evaluation of CodonMoETextCNN Variant (newly updated CodonMoE variant)** [A.8, A.10, A.11]
> > - **Evaluation on the Tc-Riboswitches Dataset (newly added dataset)** [A.14]
> > - **Additional Ablation Studies** [A.10, A.11]
> > - **Evaluation of DNA Pretrained Model Feature Effectiveness** [A.9]
> >
> > These additional experiments provide a more comprehensive evaluation of our framework, demonstrating its robustness and generalization capabilities across various models and datasets. We believe these enhancements further complement and strengthen the empirical novelty and contributions of our work.
> >
> > Thank you once again for your time and consideration.

---

> > > ### Comment · Reviewer_e4Kj · 2024-12-02
> > >
> > > Thank you for your further clarification. I am maintaining my score. For a technical paper, I would expect a more rigorous ablation study to thoroughly demonstrate its advantages and disadvantages. For an application paper, the range of downstream tasks being evaluated should be significantly broader. In my opinion, this paper falls slightly below the acceptance criteria for a top AIML conference, which is reflected in my score of 5.

---

> > > > ### Author Response · Authors · 2024-12-04
> > > >
> > > > Thank you for Reviewer e4Kj's feedback. We acknowledge the importance of comprehensive ablation studies to thoroughly demonstrate the strengths and limitations of our approach. We have conducted additional experiments comparing the CodonMoE architecture with a standard dense baseline model. Specifically, we introduced the **HyenaDNA-Densebaseline_TextCNN** model, which replaces the CodonMoE module with standard dense layers while maintaining an equivalent number of trainable parameters and identical training hyperparameters. The results are as follows:
> > > >
> > > > | Model                           | Vaccine Degradation | mRFP Expression |
> > > > |---------------------------------|---------------------|-----------------|
> > > > | HyenaDNA-Densebaseline_TextCNN  | 0.80                | 0.82            |
> > > > | HyenaDNA-CodonMoE_TextCNN       | 0.84                | 0.85            |
> > > >
> > > > These results demonstrate that the performance gains are attributable to the CodonMoE architecture itself, rather than merely an increase in parameters. This comparison is detailed in Appendix A.8, ensuring a fair and controlled ablation study.
> > > >
> > > > To address the concern regarding the breadth of downstream tasks, we have expanded our evaluation to include the **Tc-Riboswitches dataset**, as detailed in Appendix A.14. This addition complements our existing tasks targeting switching factors (vaccine degradation) and expression (mRFP expression), thereby providing a more comprehensive assessment of our framework’s generalization capabilities across diverse biological functions. The inclusion of this dataset ensures that our evaluation is representative of various categories, enhancing the robustness of our proposed method.
> > > >
> > > > Further strengthening our ablation studies, we have conducted analyses to evaluate the impact of different model components. Specifically, we examined the effect of integrating **TextCNN heads versus MLP heads** within the CodonMoE module. The results, presented in Appendices A.10 and A.11, indicate that TextCNN heads outperform MLP heads when combined with DNA pre-trained models. This underscores the importance of selecting appropriate regression heads to maximize model performance.
> > > >
> > > > We recognize the merit in exploring additional experiments, such as integrating CodonMoE with mRNA-specific models and using protein-based language models as baselines. However, these directions extend beyond the primary scope of our current research, which aims to adapt DNA-based models for RNA tasks efficiently. We plan to investigate these valuable suggestions in our future work to further validate and extend the applicability of the CodonMoE framework.
> > > >
> > > > **Summary of Additional Experiments**
> > > > In our rebuttal, we have added experiments following your suggestions. Here is a summary of the added experiments included:
> > > >
> > > > - **Additional Baseline for Ablation Study:** Comparing CodonMoE with a dense baseline (HyenaDNA-Densebaseline_TextCNN) that maintains the same number of trainable parameters and uses identical training hyperparameters.
> > > > - **Evaluation on the Tc-Riboswitches Dataset:** Broadening the range of downstream tasks to demonstrate generalization across a new category, “targeting Switching Factor,” complementing the existing categories of “targeting degradation” (vaccine degradation) and “targeting expression” (mRFP expression).
> > > > - **Additional Ablation Studies:** Further analysis of model components, including the integration of TextCNN heads, which significantly outperform MLP heads in combination with DNA pre-trained models, as shown in Appendices A.10 and A.11.
> > > > - **Evaluation of DNA Pretrained Model Feature Effectiveness:** Demonstrating the effectiveness of DNA-pretrained models on new datasets, ensuring comprehensive validation of our approach.
> > > >
> > > > **Conclusion**
> > > > We believe these enhancements strengthen our manuscript by offering a more thorough evaluation of our method’s advantages and limitations, demonstrating the robustness and versatility of our proposed framework. We hope that these more comprehensive analyses align with the high standards expected for a top AI conference and showcase the empirical novelty and contributions of our work.
> > > >
> > > > Thank you again for your feedback and consideration.

---

### Official Review · Reviewer_RPoS · 2024-11-03

**Soundness:** 3
**Presentation:** 2
**Contribution:** 2
**Rating:** 5
**Confidence:** 3

**Summary:**

The authors propose a MoE-based head for finetuning DNA LMs for RNA specific tasks.
They apply their CodonMoE head on 2 DNA LMs (Evo and Caduceus) and demonstrate improved Spearman correlation on 2 benchmarking datasets (mRFP expression  and SARS-cov-2 vaccine degradation).

**Strengths:**

The authors introduce a method for improving DNA LMs on RNA tasks in a parameter efficient way by utilizing codon structure in their model architecture. The benchmark improvements seem to be significant.

**Weaknesses:**

The authors show that CodonMoE improves performance on RNA tasks, however it is not clear that the MoE is at all necessary, and may just be adding unnecessary complexity for a 20M param head. CodonMean is a good baseline for ablation, however this baseline does not have additional training parameters unlike CodonMoE (20M parameters). The authors should include an additional baseline to specifically ablate the mixture-of-experts architecture vs. standard dense transformer layers. This baseline should have the same number of trainable parameters as CodonMoE and use exactly the same training hyper-parameters.

**Questions:**

- How is train/test split produced for each tasks. Did the authors apply sequence similarity based split, or how is train/test leakage avoided?

-Table 1 Model parameters are misleading. It does not contain the parameter count for the base models (Caduceus, etc), but does for CodonBert? This column should provide the number of params for the base mode, and the additional params for CodonMoE head. Additionally, the authors should give the exact number of parameters in M, not >80M and <20M.

---

> ### Author Response · Authors · 2024-11-22
> **Response to Reviewer RPoS (1/N)**
>
> We sincerely thank Reviewer RPoS for the feedback and will respond to the raised questions below.
>
> Q1.
> >The authors show that CodonMoE improves performance on RNA tasks, however it is not clear that the MoE is at all necessary, and may just be adding unnecessary complexity for a 20M param head. CodonMean is a good baseline for ablation, however this baseline does not have additional training parameters unlike CodonMoE (20M parameters).
>
> A1.
> We thank the reviewer for pointing out the need for clarity regarding the parameter contribution of the CodonMoE module and its necessity in improving RNA task performance.
>
> To address this concern, we clarify that the parameter count labeled as “<20M” refers to the entire framework's parameters and not solely the parameters introduced by the CodonMoE module. For instance, the HyenaDNA-CodonMoE model comprises a total of 12.7M parameters, which includes both the base HyenaDNA backbone and the CodonMoE module. This is still far fewer parameters compared to RNA-specific models like CodonBERT, which has 81.7M parameters. Despite this significantly smaller parameter footprint, HyenaDNA-CodonMoE achieves comparable or superior performance, demonstrating the module’s utility without introducing unnecessary complexity.
>
> We also note that CodonMean serves as a baseline for ablation, designed to isolate the role of RNA-specific control mechanisms in performance gains. While CodonMean does not add additional trainable parameters, it lacks the sophisticated routing mechanism of CodonMoE, which enables adaptive learning at the codon level. The observed improvements in RNA task performance when using CodonMoE are thus attributable to its ability to dynamically capture RNA-specific features, not merely to an increase in parameters.
>
> Additionally, in the table below,  to address concerns about fair comparisons, the HyenaDNA-CodonMoE_TextCNN model includes modifications to balance parameter contributions (7.5M parameters in total). Even with fewer parameters than HyenaDNA-CodonMoE, this configuration achieves performance on par with CodonBERT, demonstrating the efficacy of CodonMoE's design and its ability to enhance performance efficiently.
>
> | **Method**                                   | **Modality** | **Time Complexity** | **Model Parameters** | **Vaccine Degradation** | **mRFP Expression** |
> |:--------------------------------------------|:------------:|:--------------------:|:--------------------:|:------------------------:|:-------------------:|
> | **RNA Models**                              |              |                      |                      |                          |                     |
> | CodonBERT                                    | RNA          | quadratic            | 81.7M                | **0.77**                 | **0.85**            |
> | **DNA Models**                              |              |                      |                      |                          |                     |
> | GPN-MSA                                      | DNA          | quadratic            | 85.7M                | 0.55                     | 0.33                |
> | GPN-MSA-CodonMoE                             | DNA          | quadratic            | 161.9M               | 0.77                     | 0.79                |
> | GPN-MSA-CodonMoE_TextCNN          | DNA          | quadratic            | 115.0M               | 0.82                     | 0.81                |
> | HyenaDNA                                     | DNA          | subquadratic         | 4.1M                 | 0.69                     | 0.44                |
> | HyenaDNA-CodonMoE                             | DNA          | subquadratic         | 12.7M                | 0.81                     | 0.84                |
> | HyenaDNA-CodonMoE_TextCNN          | DNA          | subquadratic         | 7.5M                 | **0.84**                 | **0.85**            |
>
> *CodonMoE suffix indicates models enhanced with our proposed CodonMoE module. CodonMoE_TexCNN suffix indicates models enhanced with our proposed CodonMoE module with a TextCNN head. Each data set is split into training, validation, and test with a 0.7, 0.15, and 0.15 ratio. All the methods were optimized on the same data split, using the same split set as in the CodonBERT [1]. The metric is Spearman's rank Correlation.*
>
> Q2.
> >How is train/test split produced for each tasks. Did the authors apply sequence similarity based split, or how is train/test leakage avoided?
>
> A2.
> As mentioned in table above, Each data set is split into training, validation, and test with a 0.7, 0.15, and 0.15 ratio. All the methods were optimized on the same data split, using the same split set as in the CodonBERT [1].
>
> [1] Sizhen Li, Saeed Moayedpour, Ruijiang Li, Michael Bailey, Saleh Riahi, Lorenzo Kogler-Anele,
> Milad Miladi, Jacob Miner, Fabien Pertuy, Dinghai Zheng, et al. CodonBERT large language
> model for mRNA vaccines. Genome Research, 34(7):1027–1035, 2024.

---

> ### Author Response · Authors · 2024-11-22
> **Response to Reviewer RPoS (2/N)**
>
> We sincerely thank Reviewer RPoS for the feedback and will respond to the raised questions below.
>
> Q3.
>
> >-Table 1 Model parameters are misleading. It does not contain the parameter count for the base models (Caduceus, etc), but does for CodonBert? This column should provide the number of params for the base mode, and the additional params for CodonMoE head. Additionally, the authors should give the exact number of parameters in M, not >80M and <20M.
>
> A3.
> To clarify, the parameter ranges provided in the original table were included to offer a general comparison of parameter magnitudes between the different frameworks. This was intended to aid in highlighting the significant difference in parameter efficiency between DNA-based models augmented with CodonMoE and RNA-specific models like CodonBERT [1].
> However, we understand the concern that providing exact parameter counts for primary comparing models would enhance transparency and precision. These main values are indeed presented in the above table, where we report the exact parameter counts: HyenaDNA (4.1M), HyenaDNA-CodonMoE (12.7M), HyenaDNA-CodonMoE_TextCNN (7.5M) and CodonBERT (81.7M).
>
> Q4.
>
> >The authors should include an additional baseline to specifically ablate the mixture-of-experts architecture vs. standard dense transformer layers. This baseline should have the same number of trainable parameters as CodonMoE and use exactly the same training hyper-parameters.
>
> A4.
> we conducted additional experiments comparing the CodonMoE architecture with a dense baseline model using our proposed framework HyenaDNA-CodonMoE_TextCNN for fair ablation studies in the table below. The dense baseline replaces the CodonMoE architecture with standard dense layers while maintaining an equivalent number of trainable parameters and identical training hyperparameters across both models. This controlled setup isolates the specific impact of the CodonMoE architecture on model performance.
>
> The dense baseline (`HyenaDNA-Densebaseline_TextCNN`): The standard dense layer module was plugged into the HyenaDNA backbone, matching the number of trainable parameters introduced by the CodonMoE module. The dense baseline uses the same TextCNN head and training hyper-parameters as the CodonMoE-enhanced model to ensure a fair comparison.
>
> CodonMoE-enhanced Model (`HyenaDNA-CodonMoE_TextCNN`): The original CodonMoE module integrated into the HyenaDNA backbone, utilizing the TextCNN head and identical training hyper-parameters.
>
> | Model                                      | Vaccine Degradation | mRFP Expression |
> |--------------------------------------------|---------------------|-----------------|
> | HyenaDNA-Densebaseline_TextCNN | 0.80                | 0.82            |
> | HyenaDNA-CodonMoE_TextCNN  | 0.84                | 0.85            |
>
> *Each data set is split into training, validation, and test with a 0.7, 0.15, and 0.15 ratio. All the methods were optimized on the same data split, using the same split set as in the CodonBERT [1]. The metric is Spearman's rank Correlation.*

---

### Official Review · Reviewer_gygq · 2024-11-03

**Soundness:** 2
**Presentation:** 3
**Contribution:** 2
**Rating:** 6
**Confidence:** 3

**Summary:**

The authors suggest a strategy to adapt models trained on DNA data for mRNA downstream tasks. The proposed strategy uses the pre-trained language model as a backbone model to create nucleotide-level representations of the sequence. From the nucleotide-level representations codon-level representations are created and fed into the MoE module, and eventually the output representation is fed into prediction layers for the downstream tasks. The authors show for three different backbone models (HYenaDNA, Caduceus, and GPN-MSA) that the performance on the RNA downstream tasks can be improved by using their MoE adapter.

**Strengths:**

- **Novelty**: Adapting DNA models for RNA downstream tasks with an MoE approach is novel and could be interesting.
- **Relevance**:
  * The authors show that their adaption strategy improves backbone model performances on codon-level RNA downstream tasks (compared to the not fine-tuned backbone models). The presented performance values are comparable with selected baseline, i.e models trained on the RNA modality.
  * The ablation study shows that the MoE module outperforms a simple mean baseline.
- **Clarity**: The scope of this work and the author's main contribution, i.e. applying an MoE module to codon-level inputs retrieved from a DNA backbone model, is described clearly and well.
- **Reproducibility**: The code wrt the MoE module is given. (Code for training and full reproduction is assumed to be given once acceptance.)

**Weaknesses:**

- **Novelty**: The fact that Mixture of Expert models initialized from some backbone language model typically perform well is known in the general language space [1-2] and applying this principle to the DNA/RNA domain seems straight forward.
- **Clarity/Relevance**:
  * The authors missed to provide information about how many parameters were added by the MoE module? Could performance gains just arose because of the increased number of parameters? The authors should set up an experiment in which the MoE module is compared with a fine-tuning strategy for which the parameters of the prediction head is increased such that the number of parameters of both approaches match.
  * For comparison, the authors should report performance values of the backbone models trained on the RNA domain.
  * The design choice of using codon-level representations (and also the author's choice for codon-level based downstream tasks) is not explained well. How well would an naive MoE approach without codon-level representation (on nucleotide-level based downstream tasks) work?

[1] OLMoE: Open Mixture-of-Experts Language Models

[2] Upcycling Large Language Models into Mixture of Experts

**Questions:**

- What are the performance values of the backbone models trained on the RNA domain?
- How many parameters where added to the method by using the MoE module?
- Why do models trained on the DNA domain so poor? What is the intuition behind this? DNA and RNA domain seem highly correlated.

---

> ### Author Response · Authors · 2024-11-23
> **Response to Reviewer gygq (1/N)**
>
> We sincerely appreciate Reviewer gygq for the feedback and address the questions raised below.
>
> Q1.
>
> >Novelty: The fact that Mixture of Expert models initialized from some backbone language model typically perform well is known in the general language space [1-2] and applying this principle to the DNA/RNA domain seems straight forward.
>
> A1.
>
> We appreciate the reviewer’s insights regarding the general understanding of Mixture of Expert (MoE) models and their applicability in various domains. However, we would like to clarify several points:
>
> (Focus of Our Contribution) The primary focus of our work is not on the MoE design itself but on its novel adaptation to the genomic domain. While the general principles of MoE are well-known in the language modeling space, applying these principles effectively to biological sequence modeling presents unique challenges and opportunities, which our work tries to address. Our contributions lie in demonstrating the efficacy of this approach in a highly specialized domain, introducing domain-specific novel techniques, and evaluating performance on RNA-specific downstream tasks. Unlike general MoE models used in the language modeling space, we specifically designed a new Adaptive Mixture of Codon Reformative Experts (CodonMoE) tailored for genomic language modeling. This architecture leverages the unique structure and biological significance of codons, making it fundamentally different from generic MoE designs.
>
> (Universal Approximation at the Codon Level)  One of our theoretical contributions is proving that CodonMoE serves as a universal approximator at the codon level. We demonstrate that, given sufficient capacity, our proposed CodonMoE can approximate any function mapping codon sequences to target RNA properties with arbitrary precision when integrated with the pretrained backbone model. This theoretical foundation underscores the distinctiveness of our approach compared to existing methods.
>
> (Timeline of Relevant Work) We acknowledge the existence of the referenced paper [1] and [2] (posted on arxiv on 2024.10.10), which seems related to our work and we will cite these in our future manuscript. However, we would like to clarify that this paper [1] was public after the submission deadline of our manuscript. Thus, our work was developed independently and without the influence of this more recent contribution. Furthermore, while the referenced paper might overlap conceptually with our approach, our work focuses on distinct aspects, including domain-specific design of the general MoE, task evaluation, and interpretability in the genomic modeling context.
>
> We hope this clarifies the novelty and the independent development of our contributions. Thank you for your feedback!
>
> [1] Niklas Muennighoff, Luca Soldaini, Dirk Groeneveld, Kyle Lo, Jacob Morrison, Sewon Min, Weijia Shi, Pete Walsh, Oyvind Tafjord, Nathan Lambert, et al. Olmoe: Open mixture-of-experts language models. arXiv preprint arXiv:2409.02060, 2024.
>
> [2] Ethan He, Abhinav Khattar, Ryan Prenger, Vijay Korthikanti, Zijie Yan, Tong Liu, Shiqing Fan, Ashwath Aithal, Mohammad Shoeybi, and Bryan Catanzaro. Upcycling large language models into mixture of experts. arXiv preprint arXiv:2410.07524, 2024.

---

> ### Author Response · Authors · 2024-11-23
> **Response to Reviewer gygq (2/N)**
>
> Q2.
> >Clarity/Relevance: The authors missed to provide information about how many parameters were added by the MoE module? Could performance gains just arose because of the increased number of parameters?
>
> A2.
> We appreciate the reviewer’s concerns about the parameter of the CodonMoE module and their impact on performance gains. We provide additional clarity below to address these points.
>
> The CodonMoE module adds a modest number of parameters to our framework. For instance, the **HyenaDNA-CodonMoE** model has **12.7M parameters**, which is significantly fewer than RNA-specific models like **CodonBERT** with **81.7M parameters**. Despite the smaller size, HyenaDNA-CodonMoE achieves performance comparable to or better than CodonBERT on both mRFP expression and vaccine degradation tasks. Additionally, the **HyenaDNA-CodonMoE_TextCNN** variant, with **7.5M parameters**, matches or exceeds CodonBERT’s performance, demonstrating that the improvements are due to the specialized design of CodonMoE rather than just an increase in parameters. These results demonstrate that **CodonMoE's architectural enhancements** provide significant performance gains beyond what is achieved by merely increasing the number of parameters.
>
> Additionally, in the table below,  to address concerns about fair comparisons, the HyenaDNA-CodonMoE_TextCNN model includes modifications to balance parameter contributions. Even with fewer parameters than HyenaDNA-CodonMoE, this configuration achieves performance on par with CodonBERT, demonstrating the efficacy of CodonMoE's design and its ability to enhance performance efficiently.
>
> | Method                          | Modality | Time Complexity | Base Parameters | CodonMoE Parameters | Total Parameters | Vaccine Degradation | mRFP Expression |
> |---------------------------------|----------|------------------|------------------|----------------------|------------------|---------------------|-----------------|
> | **RNA Models**                                                                                                                           |
> | CodonBERT                      | RNA      | quadratic        | 81.7M           | -    | 81.7M           | **0.77**            | **0.85**        |
> | **DNA Models**                                                                                                                           |
> | GPN-MSA                        | DNA      | quadratic        | 85.7M           | -                   | 85.7M           | 0.55                | 0.33            |
> | GPN-MSA-CodonMoE               | DNA      | quadratic        | 85.7M           | 76.2M               | 161.9M          | 0.77                | 0.79            |
> | GPN-MSA-CodonMoE_TextCNN       | DNA      | quadratic        | 85.7M           | 29.3M               | 115.0M          | 0.82                | 0.81            |
> | HyenaDNA                       | DNA      | subquadratic     | 4.1M            | -                   | 4.1M            | 0.69                | 0.44            |
> | HyenaDNA-CodonMoE              | DNA      | subquadratic     | 4.1M            | 8.6M                | 12.7M           | 0.81                | 0.84            |
> | HyenaDNA-CodonMoE_TextCNN      | DNA      | subquadratic     | 4.1M            | 3.4M                | 7.5M            | **0.84**            | **0.85**        |
>
>
> *The CodonMoE_TextCNN suffix denotes  the CodonMoE module where the final dense layers have been replaced with a TextCNN head. Performance is measured using Spearman's rank Correlation.* Data splits: 70% train, 15% validation, 15% test (same as CodonBERT [3]).
>
>
> Q3.
>
> > The authors should set up an experiment in which the MoE module is compared with a fine-tuning strategy for which the parameters of the prediction head is increased such that the number of parameters of both approaches match.
>
> A3.
> To determine if the performance gains are solely due to an increased number of parameters, we also conducted ablation experiments comparing CodonMoE with a dense baseline as presented in the table below. The dense baseline (HyenaDNA-Densebaseline_TextCNN) replaces the CodonMoE module with standard dense layers, ensuring both models have the same number of trainable parameters and identical training hyperparameters.
> | Model                                      | Vaccine Degradation | mRFP Expression |
> |--------------------------------------------|---------------------|-----------------|
> | HyenaDNA-Densebaseline_TextCNN | 0.80                | 0.82            |
> | HyenaDNA-CodonMoE_TextCNN  | 0.84                | 0.85            |
>
> *Data splits: 70% train, 15% validation, 15% test (same as CodonBERT [3]).
> Metric: Spearman's rank Correlation.*
>
> [3] Sizhen Li, Saeed Moayedpour, Ruijiang Li, Michael Bailey, Saleh Riahi, Lorenzo Kogler-Anele, Milad Miladi, Jacob Miner, Fabien Pertuy, Dinghai Zheng, et al. CodonBERT large language model for mRNA vaccines. Genome Research, 34(7):1027–1035, 2024.

---

> ### Author Response · Authors · 2024-11-23
> **Response to Reviewer gygq (3/N)**
>
> Q4.
> >For comparison, the authors should report performance values of the backbone models trained on the RNA domain. What are the performance values of the backbone models trained on the RNA domain?
>
> A4.
> We appreciate the reviewer’s suggesting we report performance metrics for backbone models trained on the RNA domain.
>
> However, our primary goal with CodonMoE is to enhance DNA models for RNA tasks, thereby avoiding the high computational costs of training separate RNA-specific models like CodonBERT. This design helps reduce the cost of pretraining separate foundation models for both DNA and RNA domains, and maintain high performance efficiently.
>
> By focusing on DNA models enhanced with CodonMoE, we demonstrate a cost-efficient solution that maintains high performance on RNA tasks without the need for a separately pretrained RNA-specific backbone.
>
> Q5.
> >The design choice of using codon-level representations (and also the author's choice for codon-level based downstream tasks) is not explained well. How well would an naive MoE approach without codon-level representation (on nucleotide-level based downstream tasks) work?
>
> A5.We appreciate the reviewer’s question regarding the rationale behind using codon-level representations and codon-based downstream tasks. We address this below:
>
> 1.Codons represent the fundamental biological units of genetic coding, where each codon corresponds to a specific amino acid. By focusing on codon-level representations, we directly incorporate biologically meaningful information into the model. This approach allows CodonMoE to capture higher-order patterns and relationships that are essential for genomic tasks, which might be lost in nucleotide-level representations.
>
> 2.Our work focuses on general mRNA tasks without distinguishing between codon-level or nucleotide-level downstream tasks. Codon-aware representations provide a robust foundation for these tasks by incorporating essential biological guidance without introducing unnecessary parameters.
>
> 3.Using codon-level representations in MoE reduces the number of sequence tokens compared to nucleotide-level representations. Codon-aware representations integrate essential biological information with fewer parameters, making them preferable to nucleotide-level approaches that lack biologically guided structures. Codon-level representations enable CodonMoE to leverage natural codon structures, better aligning with biological processes and enhancing task performance.
>
> 4.We will incorporate better explanation into the manuscript to clarify our design choices and provide additional context for the use of codon-level representations.
>
> Q6.
> >Why do models trained on the DNA domain so poor? What is the intuition behind this? DNA and RNA domain seem highly correlated.
>
> A6.We appreciate the reviewer’s question about the intuition behind the performance gap of DNA-trained models on RNA tasks. While DNA and RNA domains are indeed highly correlated, there are important distinctions that explain this performance difference:
>
> 1.DNA models are specifically trained on DNA-related tasks, optimizing for patterns, structures, and features that are distinct to DNA sequences. RNA tasks often involve additional complexities, such as secondary structures, splicing events, and regulatory roles, which are not explicitly modeled or emphasized in DNA-specific training. This specialization limits the ability of DNA-trained models to generalize effectively to RNA-specific tasks.
>
> 2.While DNA and RNA are biochemically related, they serve distinct biological roles. DNA primarily acts as a storage medium for genetic information, while RNA is involved in dynamic processes such as transcription, translation, and regulation. These functional differences translate into divergent sequence features and contextual dependencies, making it challenging for DNA-trained models to directly perform well in RNA tasks. Codons are one such (very important) sequence feature of RNA, that while interpretable from the DNA, are best modeled directly when dealing with RNA.
>
> 3.Many RNA tasks, such as predicting RNA secondary structure or identifying splicing variants, rely on domain-specific features and representations. These features are not explicitly modeled or prioritized in DNA-trained models, leading to suboptimal performance on RNA tasks.
>
> 4.Recognizing these challenges, our approach focuses on leveraging the strengths of DNA-trained models by introducing CodonMoE. This enhancement allows DNA models to better adapt to RNA tasks by designing codon-aware MoE.
>
> We hope this explanation clarifies the observed performance differences and the underlying intuition. Thank you for raising this important point!

---

> ### Comment · Reviewer_gygq · 2024-11-28
> **Reviewer's answer to the Authors**
>
> Dear Authors,
>
> thank your for your detailed answers. Please find my follow-up comments below:
>
> **Novelty**: Indeed the authors are right, suggested related work should be considered concurrent work.
>
> **Q2/A2**: Thanks for clarification.
>
> **Q3/A3**: This experiment is very relevant since it shows that performance gains do not stem just from an increased parameter count.
>
> **Q4/A4**:  Still, reporting performance metrics for backbone models trained on the RNA domain would have been interesting.
>
> **Q5/A5/Q6/A6**: Thank you for your detailed answers. The manuscript would benefit from adding this information as a motivation for the chosen approach.
>
> Since I think the quality of the proposed work improved and my novelty concerns with respect to related work (Q1/A1) became obsolete, I decided to increase my score to 6.

---

> > ### Author Response · Authors · 2024-11-29
> >
> > Dear Reviewer gygq,
> >
> > Thank you for your thoughtful feedback and for increasing your score. We appreciate your recognition of the improvements made.
> >
> > Novelty: We are pleased that addressing concurrent related work has clarified our contributions.
> >
> > Q2/A2: Thank you for acknowledging our clarification.
> >
> > Q3/A3: We are glad that the additional experiment demonstrated the robustness of our method beyond parameter count.
> >
> > Q4/A4: We understand the value of reporting RNA-trained backbone metrics and will consider this in future work.
> >
> > Q5/A5/Q6/A6: We agree that adding motivational information strengthens our manuscript and will incorporate these details.
> >
> > Thank you again for your support!

---

### Author Response · Authors · 2024-11-25
**General Response**

We sincerely thank all the reviewers for their time and insightful feedback.

We are encouraged by the positive reception of our work, with reviewers recognizing it as both **novel and meaningful** (gygq, e4Kj) in genomic language model field. The appreciation for our **CodonMoE framework's adaptability, parameter efficiency, and performance improvements** ( gygq, RPoS, e4Kj) was particularly gratifying.

The importance of our work lies in its potential to unify DNA and RNA modeling efforts, thereby reducing computational costs and resource requirements. By introducing the Adaptive Mixture of Codon Reformative Experts (CodonMoE), we address the critical challenge of maintaining separate models for DNA and RNA tasks, offering a scalable solution.

**Key Contributions:**

- **Adaptive Mixture of Experts:** Introducing CodonMoE, a novel module that effectively adapts DNA language models for RNA-based predictive tasks.
- **Parameter Efficiency:** Demonstrating that CodonMoE-enhanced HyenaDNA achieves comparable or better performance with significantly fewer parameters compared to RNA-specific models on three mRNA tasks.
- **Comprehensive Evaluation:** Providing extensive experimental results across various DNA-based backbones and demonstrating substantial improvements compared with DNA backbones as well as comparable or superior performance to state-of-the-art RNA models.
- **Theoretical Foundation:** Proving that CodonMoE serves as a universal approximator at the codon level, ensuring robust performance across diverse RNA tasks.
- **Scalability and Practical Impact:** Offering a plug-and-play solution that can be easily integrated into existing DNA models, promoting widespread adoption and reducing the computational burden of maintaining separate models.

**Additional Experiments**

In this rebuttal, we have added more supporting experiments following reviewers' suggestions. Here we summarize the new experiments, which have been updated in the manuscript from Appendix 7 to Appendix 14 (colored blue):

- **Updated Main Table with Detailed Parameters and Enhanced Models** [A.7]
- **Introduction and Evaluation of CodonMoETextCNN Variant** (newly updated CodonMoE variant) [A.8, A.10, A.11]
- **Evaluation on the Tc-Riboswitches Dataset** (newly added dataset) [A.14]
- **Additional Ablation Studies** [A.10, A.11]
- **Updated Evaluation of DNA Pretrained Model Feature Effectiveness** [A.9]

We hope these new experiments further complement and strengthen the empirical novelty and contributions of our work.

We are committed to thoroughly addressing each reviewer's comments and suggestions to improve our manuscript. Detailed responses to specific feedback are provided below, incorporating additional experiments and clarifications as needed.

We reiterate our dedication to refining our work based on the valuable input received and believe these enhancements will further strengthen the impact of our contributions.

Thank you once again for your valuable feedback and consideration.

---

### Meta-Review · Area_Chair_wwVr · 2025-01-02

**Metareview:**

The paper presents a MoE method for adapting models trained on DNA data for downstream mRNA tasks. The task is undoubtedly important, and the use of MoE models in this setting is also new. However, there were nontrivial concerns about the experimental design -- specifically, about the use of a limited number of downstream tasks and the quality of the ablation study. The new results that the authors added during the rebuttal period partly, but not entirely, address these concerns. As a result, I am recommending rejection this time around. I encourage the authors to incorporate the feedback in the reviews and submit to a different deadline.

**Additional Comments On Reviewer Discussion:**

The authors added new results during the rebuttal period, and there was a spirited back-and-forth between the authors and the primary critical reviewer. In the end, the authors did not fully convince the reviewer.

---

### Decision · Program_Chairs · 2025-01-22

Reject